# Manipulation of RNA polymerase III by Herpes Simplex Virus-1

Sarah E. Dremel [1,5], Frances L. Sivrich[1], Jessica M. Tucker[2,6], Britt A. Glaunsinger [2,3,4] & Neal A. DeLuca [1✉]

RNA polymerase III (Pol III) transcribes noncoding RNA, including transfer RNA (tRNA), and is commonly targeted during cancer and viral infection. We find that Herpes Simplex Virus-1 (HSV-1) stimulates tRNA expression 10-fold. Perturbation of host tRNA synthesis requires nuclear viral entry, but not synthesis of specific viral transcripts. tRNA with a specific codon bias were not targeted—rather increased transcription was observed from euchromatic, actively transcribed loci. tRNA upregulation is linked to unique crosstalk between the Pol II and III transcriptional machinery. While viral infection results in depletion of Pol II on host mRNA promoters, we find that Pol II binding to tRNA loci increases. Finally, we report Pol III and associated factors bind the viral genome, which suggests a previously unrecognized role in HSV-1 gene expression. These findings provide insight into mechanisms by which HSV-1 alters the host nuclear environment, shifting key processes in favor of the pathogen.

[1] Department of Microbiology and Molecular Genetics, University of Pittsburgh School of Medicine, Pittsburgh, PA, USA. [2] Department of Plant and Microbial Biology, University of California Berkeley, Berkeley, CA, USA. [3] Department of Molecular and Cell Biology, University of California Berkeley, Berkeley, CA, USA. [4] Howard Hughes Medical Institute, Mount Shasta, CA, USA. [5] Present address: HIV and AIDS Malignancy Branch, Center for Cancer Research, National Cancer Institute, National Institutes of Health, Bethesda, MD, USA. [6] Present address: Department of Microbiology and Immunology, Carver College of Medicine, University of Iowa, Iowa City, IA, USA. ✉email: ndeluca@pitt.edu

Herpes Simplex Virus-1 (HSV-1) is a ubiquitous human pathogen which most commonly causes recurrent lesions of the oral and genital mucosa. The virus is associated with a wide range of additional pathologies—including herpes keratitis, herpetic whitlow, and encephalitis, to name a few—representative of the large range of cells permissive to replication. Similar to other herpesviruses, HSV-1 establishes a latent reservoir in the peripheral nervous system and reactivates to cause disease in response to various physiological stimuli.

HSV-1 replicates and assembles almost entirely within the host nucleus, reprogramming the host transcriptional machinery to prioritize expression of ~90 viral messenger RNAs (mRNAs). These genes are transcribed in a temporally coordinated sequence, such that their protein products are expressed at the appropriate time in the life cycle of the virus[1]. Immediate early (IE or α) gene products enable the efficient expression of early (E or β) and late (L or γ) genes. The protein products of E genes are mostly involved in DNA replication. DNA replication and IE proteins enable the efficient transcription of L genes, which encode the structural components of the virus. DNA replication licenses L promoters, enabling the binding of core Pol II transcription factors, thus activating the initiation of L transcription[2]. Productive HSV-1 infection is incredibly rapid, causing a single infected cell to produce progeny between 4 and 6 hours (h) postinfection, culminating in ~1000 infectious progeny within 18 h. Considering the rapid HSV-1 life cycle, it is perhaps unsurprising that within 6 h of infection, viral transcripts rise to 50% of the total mRNA within a host cell. The corresponding decrease in host transcripts is facilitated by two mechanisms (i) VHS-mediated mRNA decay[3] and (ii) ICP4-DNA mediated decrease of Pol II on mRNA promoters[4–7].

Transcriptional studies of HSV center on mRNA and RNA Polymerase II (Pol II), and thus largely overlook the potential contributions of other DNA-dependent RNA Polymerases, namely Pol I and Pol III. Pol I transcribes a single multi-copy transcript, 45 S, which is spliced and processed to produce 5.8 S, 18 S, and 28 S ribosomal RNA (rRNA). Pol III transcribes noncoding RNAs, including 5 S rRNA, transfer RNA (tRNA), Alu elements, 7SL, 7SK, U6, and select microRNA (miRNA). Pol III also transcribes noncoding RNAs (ncRNA) for various DNA viruses, including: VAI and VAII RNAs of adenovirus[8], EBER1 and EBER2 of Epstein Barr Virus[9] and the tRNA-miRNA encoding RNAs (TMERs) of murine gammaherpesvirus-68 (MHV68)[10].

In this study, we explore HSV-1 alteration of the host Pol III transcription landscape. Pol III transcription requires its own unique set of general transcription factors, and the basal promoter requirements have been classified into three different promoter types[11]. Type I promoters include 5 S rRNA genes, and use the TFIIIB complex (BDP1, TBP, BRF1), TFIIIC complex (GTF3C1, GTF3C2, GTF3C3, GTF3C4, GTF3C5, GTF3C6), and TFIIIA (GTF3A). Type II promoters include tRNAs and require the TFIIIB and TFIIIC complexes. Type III promoters include tRNA-Sec, U6 and 7SK and use distal enhancers including STAF, OCT1, SNAP as well as a distinct TFIIIB complex (BDP1, TBP, BRF2). Each of these promoters consists of distinct combinations of internal cis-acting sites (A, B, C, IE). To add another layer of complexity, recent studies have demonstrated crosstalk between Pol II and Pol III. Highly expressed Pol III transcripts are located in regions of open chromatin adjacent to Pol II promoters[12]. Additionally, Pol II binding is observed at highly expressed Pol III promoters[13], and Pol III occupancy frequently scales with nearby levels of Pol II[11,14].

Viruses have evolved unique mechanisms to invade hosts, alter cellular pathways, and redirect host factors for viral processes. A number of viruses, including adenovirus[15,16], polyomavirus (SV40)[17], Epstein Barr Virus[18], MHV68[19], and HIV[20], have been shown to increase Pol III transcription in infected cells. Adenovirus, SV40, and Epstein Barr Virus employ a viral regulatory protein (E1A, E1B, T-antigen, and EBNA1) to mediate increase of Pol III transcription factor abundance. HSV-1 has previously been shown to induce the Pol III type II transcript, Alu repeat units[21,22]. How HSV-1 affects other Pol III-dependent transcripts and the mechanism behind Alu upregulation is yet unknown.

Herein we report a comprehensive characterization of how HSV-1 alters the Pol III transcriptional landscape, ultimately increasing the pool of tRNA available during productive infection. Changes in tRNA levels coincide with increased Pol II recruitment at these loci. This discovery is at odds with the general environment of Pol II depletion from host mRNA promoters which contributes to host transcriptional shut off[4–7]. We also report, for the first time, recruitment of Pol III to the HSV-1 genome. Whether this binding event results in production of a Pol III-dependent transcript is still unknown. However, these results suggest a role for Pol III in promoting and regulating HSV-1 transcription.

## Results

**Impact of HSV-1 productive infection on Pol I and III transcripts.** We began by assessing changes in noncoding RNA species, particularly those expressed by Pol I and III. Human diploid fibroblast (MRC5) cells were mock-infected, infected with ΔICP0/4/22/27/47 (d109), ΔICP4 (n12), or wildtype HSV-1. During infection with d109, the virus enters the nucleus but fails to synthesize any nascent viral proteins or viral genomes; however, it robustly stimulates a cGAS-mediated innate immune response[23]. ΔICP4 infection overproduces IE transcripts but is deficient in the synthesis of early (E) and late (L) viral proteins, nascent viral genomes, and viral progeny[24]. We observed little change in Pol I transcripts 18 S rRNA, 45 S pre-rRNA, or 5.8 S rRNA using RT-qPCR (Fig. 1A) and Northern blot analysis (Fig. 1B, Supplementary Fig. 1A). Pol III type I, II, and III transcripts were affected disparately (Fig. 1A, B, Supplementary Fig. 1). Type I transcript 5 S rRNA was unaltered by infection, while type II transcripts were strongly upregulated. Infection with ΔICP4 and wildtype HSV-1 increased the levels of type II transcripts from 2 to 16-fold (Fig. 1A–C). This upregulation was most drastic for pre-tRNA, and specific to tRNA with type II promoters as we did not observe upregulation for a tRNA encoded by a type III promoter, namely tRNA selenocysteine (tRNA Sec). We observed an increase—albeit modest 3 to 1.5-fold—in mature tRNA (Fig. 1C, Supplementary Fig. 1B). Type III transcripts U6 snRNA, tRNA-sec, and 7SK decreased minimally during infection (Fig. 1A, B, Supplementary Fig. 1A). ΔICP4 infection phenocopied tRNA transcriptional changes in wild-type infection, indicating that synthesis of E and L viral proteins, nascent viral genomes, or viral progeny was not required for alteration of the Pol III transcriptional landscape. In contrast, ΔICP0/4/22/27/47 infection had no impact on the tRNA transcriptional landscape, indicating viral nuclear entry or the induction of an innate immune response is not sufficient to induce upregulation. These results demonstrate HSV-1 selectively targets and upregulates Pol III type II transcripts.

**Alteration of the tRNA landscape by HSV-1 productive infection.** tRNA contain a high density of noncanonical nucleotides and base modifications which can impede processivity of reverse transcription, a step required in most sequencing techniques. The technique, DM-tRNA-Seq, abrogates these issues by removing strong-stop modifications with a demethylase enzyme (AlkB) and amplifying full-length cDNA products using a highly

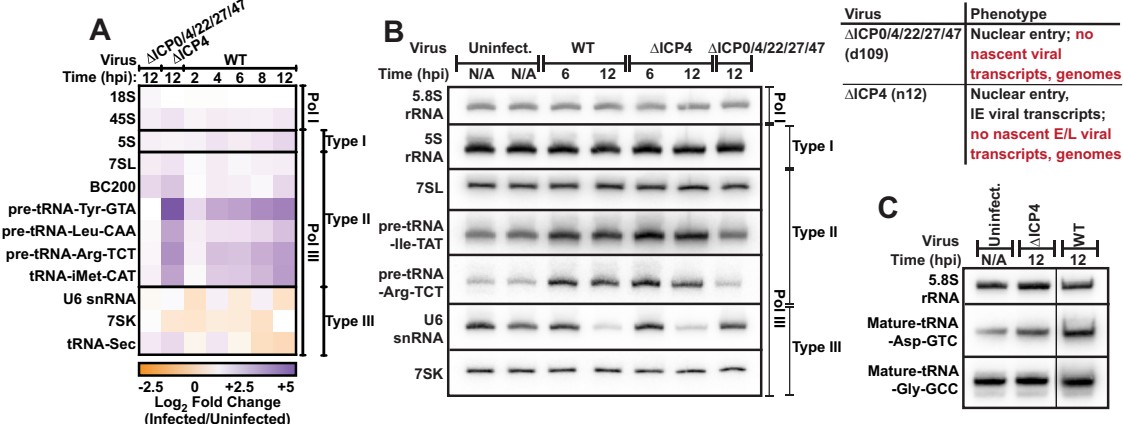

**Fig. 1 Impact of HSV-1 productive infection on Pol I and III transcripts.** Human fibroblast cells were mock-infected or infected with ΔICP0/4/22/27/47 (d109), ΔICP4 (n12), or wild-type HSV-1. RNA was isolated at indicated times, and **A** qPCR or **B**, **C** Northern blots were used to assess transcript abundance. **A** Data is the average of biological triplicate experiments, and values are average log$_2$ fold change of infected over uninfected samples. cDNA copy number was calculated using genomic DNA standard curves and as a function of total RNA used for reverse transcription. **B**, **C** Images are representative Northern blots from at least two biological replicates, data is quantified in Supplementary Fig. 1.

processive thermostable group II intron containing reverse transcriptase[25]. We performed four replicates of DM-tRNA-Seq on primary human fibroblasts mock-infected or infected with ΔICP4 (n12) or wildtype (KOS) HSV-1 for 12 h. DM-tRNA-Seq enables sequencing of full length tRNA reads, allowing for discrimination of pre- and mature-tRNA species using features including unspliced introns, leading and lagging sequences, and 3' CCA tails. We further delineated tRNA expression by those encoded from the nuclear or mitochondrial (MT) host genome.

We found that the total amount of nuclear-encoded tRNA increased 2-fold after infection with ΔICP4 or wildtype HSV-1. Pre-tRNA species were more affected than mature-tRNA, increasing 4 and 1.5-fold, respectively (Fig. 2A). Nuclear-encoded tRNA expression was altered similarly between wildtype and ΔICP4 infection (Fig. 2A–C, Supplementary Fig. 2), indicating only early viral life cycle events were required for the phenotype. In contrast, MT-encoded tRNA decreased ~4-fold only in wildtype HSV-1 infection (Fig. 2A, Supplementary Fig. 2C). We expected this decrease as HSV-1 degrades the host mitochondrial genome during productive infection[26]. MT-tRNA changed minimally in ΔICP4 infection likely due to the absence of UL12 expression, which is the viral endonuclease responsible for MT-genome degradation.

Since pre-tRNAs were more upregulated than the mature form we hypothesized that infection increased nascent transcription of nuclear-encoded tRNA. To assess this, we pulsed in 4-thiouridine (4SU) for 15 min at various stages of infection before isolating RNA and performing 4SU-Seq. tRNA are very stable ncRNAs, with half-lives of about 3 days. Using 4SU-Seq we were able to assess how tRNA transcriptional output shifts, without being confounded by tRNA made prior to infection. 4SU-Seq lacks the technical steps which ensure reverse transcription processivity, thus our data analysis has the caveat that we cannot discriminate with high confidence among degenerate isodecoders and pre- or mature-tRNA species. Based on our 4SU-Seq data sampling a 15 min pulse of nascent RNA, we expect a bulk of reads assigned to mature-tRNA species to in fact be truncated fragments from premature-tRNA.

Echoing our prior results, we observed that nuclear-encoded tRNA increased and MT-encoded tRNA decreased after infection by 4SU-Seq (Fig. 2D). Nascent nuclear tRNA levels were increased ~10-fold at 12 hpi (Fig. 2D). In Fig. 2E we show data for the top differentially expressed (DE) mature (p-value < 0.05,

log$_2$ fold change wildtype HSV-1/uninfected >0.5) and pre- (p-value < 0.05, log$_2$ fold change wildtype HSV-1/uninfected > 2) tRNA species in the DM-tRNA-Seq dataset (Supplementary Data 1). We observed an increase in nascent tRNA detected for these DE mature- and pre-tRNAs (Fig. 2E).

We next assessed whether there was enrichment for select isodecoders among the tRNA species altered by infection (Supplementary Fig. 3). The HSV-1 genome is unusually GC-rich, ~68%, which means the viral coding sequence relies more heavily on tRNA with GC-rich anticodons (Supplementary Fig. 3B). Our breakdown of DE tRNA found that the isodecoders altered did not target only GC-rich species (Supplementary Fig. 3). Generally, altered mature and pre-tRNA loci isodecoders appeared random (Supplementary Fig. 3C). Additionally, we did not observe a shift in the identity of tRNA expressed, as silenced tRNA loci were not suddenly expressed or vice versa (Supplementary Fig. 3C). Further study of the actual codon usage truly sampled during infection would need to be determined, this was merely a theoretical codon usage assuming each HSV-1 protein was synthesized in the same amount.

Thus far we have focused on upregulated tRNA, however there was a smaller subset of tRNA (n = 32 pre-tRNA, n = 10 mature tRNA) downregulated by HSV-1 infection (Fig. 2B, C). We evaluated the genomic position of up- and down-regulated tRNA and found that downregulated tRNA loci were located in close proximity to Pol II gene promoters (Supplementary Fig. 4). ~50% of downregulated tRNA loci were located within 3 kbp of a protein-coding gene promoter, whereas only ~15% of upregulated tRNA loci or all tRNA loci were promoter-adjacent (Supplementary Fig. 4D). These data suggest downregulated tRNA targets may be linked to the absence of host transcription at adjacent mRNA promoters. Based on these results it is unlikely that tRNA upregulation is due to Pol II-transcriptional interference.

**Defining HSV-1 processes required for tRNA upregulation.** Our results in Fig. 1 limit the viral processes which may be critical for tRNA upregulation, including: i. immediate early (IE) proteins other than ICP4, ii. viral ncRNA, or iii. amount, but not identity, of viral transcripts. To discriminate between these hypotheses, we infected MRC5 cells with various HSV-1 mutants and assessed tRNA abundance by Northern Blot (Fig. 3). Viruses used include: ΔICP4 (n12), ΔICP27 (5dl1.2), ΔICP0 (n212), ΔICP22 (n199), ΔICP0/4/22/27/47 (d109), ΔUL30 (hp66) or wildtype HSV-1.

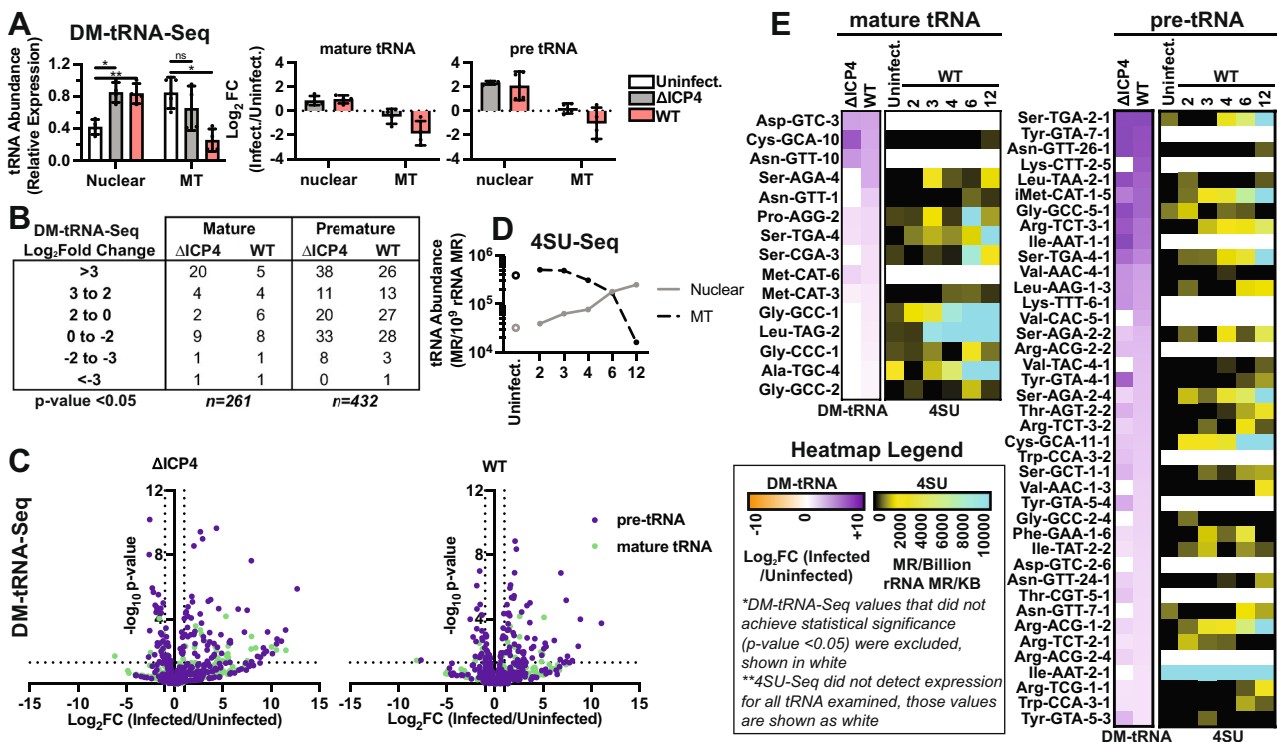

**Fig. 2 Alteration of the tRNA landscape by HSV-1 Productive Infection.** We performed DM-tRNA-Seq on human fibroblasts that were mock-infected or infected with ΔICP4 (n12) or wildtype HSV-1 (KOS) for 12 h. Data was normalized to an internal spike-in control and the size in kilobase pairs (KB) of each tRNA (mapped reads per million spike-in reads per KB, MR/MSI/KB). For 4SU-Seq, human fibroblasts were mock-infected or infected with wildtype HSV-1 and RNA was isolated at 2, 3, 4, 6, or 12 hpi. 4SU was pulsed in for 15 min at the time (hpi) indicated to label nascent RNA. Data was normalized to rRNA and the size of each tRNA (mapped reads per billion rRNA mapped reads per KB, $MR/10^9$ rRNA MR/KB). **A** Uniquely mapped DM-tRNA-Seq reads normalized to spike-in controls and delineated as those encoded by the nuclear or mitochondrial (MT) genome. Values are relative to the sample set max. Statistical values were generated from a paired $t$-test, where * indicates a $p$-value of < 0.05, or ** < 0.01. Data bars are the average, error bars are standard deviation and individual data points represent experimental replicates ($n = 4$). **B, C** Differential expression analysis (edgeR, $p$-values adjusted for multiple testing by Benjamini-Hochberg) for DM-tRNA-Seq data, requiring a $p$-value threshold of 0.05 to achieve significance. Values are $\log_2$ fold change (infected/uninfected) versus the false-discovery rate (FDR) $p$-value. **D** Uniquely mapped 4SU-Seq reads normalized to rRNA and delineated as those encoded by the nuclear or mitochondrial (MT) genome. **E** Heatmaps for upregulated mature- and pre-tRNA. Differentially expressed (DE) mature-tRNA were defined using our DM-tRNA-Seq data (analysis described in **C**) as $p$-value < 0.05, $\log_2$ fold change (HSV-1/uninfected) > 0.5. DE pre-tRNA were defined as $p$-value < 0.05, $\log_2$ fold change (HSV-1/uninfected) > 2. DM-tRNA-Seq data was plotted as $\log_2$ fold change (WT HSV-1 infected/uninfected); data with a $p$-value >0.05 was not plotted and is white in the heatmaps. 4SU-Seq data was plotted as MR/Billion rRNA MR/KB, any tRNA not detected in our data were plotted as white in the heatmaps.

ICP4, ICP27, ICP0, and ICP22 are viral IE proteins with various roles in promoting viral gene expression. UL30 is the viral DNA polymerase required for genome replication. Single-deletion mutants for the viral IE proteins—ICP4, ICP27, ICP22, ICP0—induced pre-tRNA indicating that these proteins are not themselves responsible for tRNA upregulation (Fig. 3A, Supplementary Fig. 5A). Surprisingly, infection with ΔUL30, a mutant defective for genome replication, did not induce host tRNA (Fig. 3A, Supplementary Fig. 5A). This result was phenocopied in cells treated with specific inhibitors of HSV-1 genome replication (phosphonoacetic acid and acyclovir) and infected with wildtype HSV-1 (Fig. 3B, Supplementary Fig. 5B). Since other mutants in the panel were also defective for viral genome replication, this life stage could not be responsible for induction of host tRNA (Fig. 3C). Results for other mutants in the panel confirmed that synthesis of viral L proteins, genome replication and virion assembly were not processes required for induction (Figs. 1, 3A–C).

We performed Ribominus-RNA-Seq on these samples to assess differentially expressed viral genes (Fig. 3D–F). We added ERCC spike-in controls to total RNA prior to ribosomal RNA depletion allowing us to normalize expression relative to rRNA which

remain steady during HSV-1 productive infection. This method ensures a quantitative comparison of samples with varying levels of host shut off. We identified transcriptional changes within the genes for ICP0, ICP47, ICP4, RL1, and LAT as possible candidates responsible for the differential tRNA phenotype in ΔUL30 (FC > 0 in all other viruses) (Fig. 3D). These regions of the genome also contain viral ncRNA transcripts: Latency-Associated Transcript (LAT), miRNAs, and L/STs. We used genomic lesion mutants (d120, d92, d99, R3616 F-ΔICP47), rather than nonsense mutants, to clarify our findings. Consistent with Fig. 3A, we found that the coding regions of ICP0 and ICP4 were not required for tRNA upregulation (Supplementary Fig. 6). Additionally, we found mutants with lesions for ICP47, RL1 or LAT induced host tRNA species (Supplementary Fig. 6). Taken together we concluded that ICP0, ICP47, ICP4, RL1, LAT, and miRNAs encoded within ICP0 or ICP4 were not responsible for tRNA upregulation.

This led us to our final hypothesis, in which the amount of viral transcripts—but not the identity of those expressed—is critical for tRNA upregulation. As expected, we observed robust viral transcription in ΔICP0 and wild-type HSV-1 infection (Fig. 3D–F). ICP4, ICP27, and ICP22 all function to promote

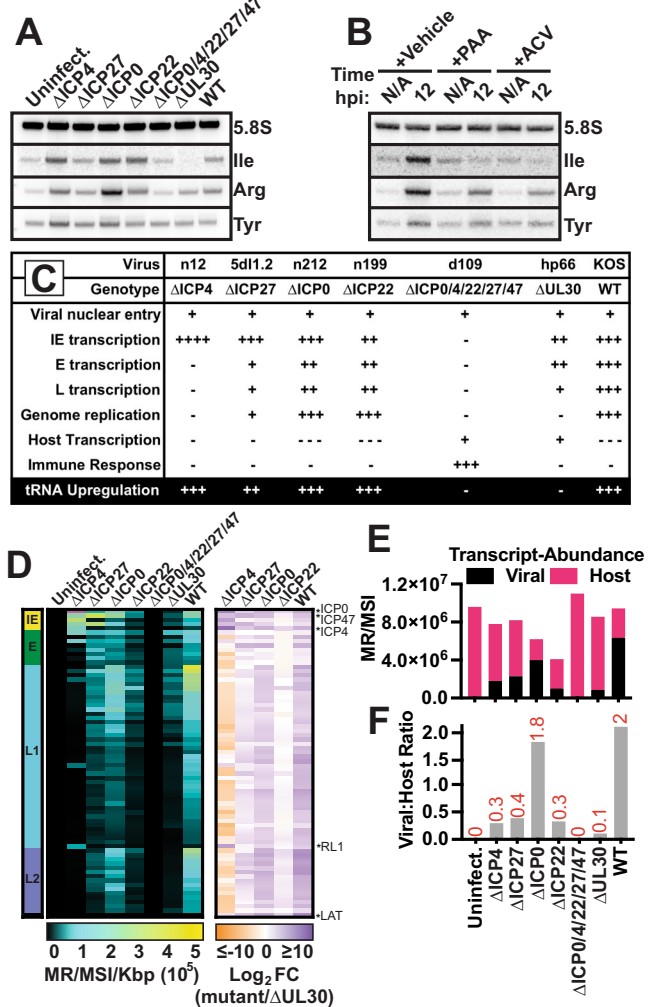

| Virus | n12 | 5dl1.2 | n212 | n199 | d109 | hp66 | KOS |
|---|---|---|---|---|---|---|---|
| Genotype | ΔICP4 | ΔICP27 | ΔICP0 | ΔICP22 | ΔICP0/4/22/27/47 | ΔUL30 | WT |
| Viral nuclear entry | + | + | + | + | + | + | + |
| IE transcription | ++++ | +++ | +++ | ++ | - | ++ | +++ |
| E transcription | - | + | ++ | ++ | - | ++ | +++ |
| L transcription | - | + | ++ | ++ | - | + | +++ |
| Genome replication | - | + | +++ | +++ | - | - | +++ |
| Host Transcription | - | - | - - - | - - - | + | + | - - - |
| Immune Response | - | - | - | - | +++ | - | - |
| tRNA Upregulation | +++ | ++ | +++ | +++ | - | - | +++ |

**Fig. 3 Defining HSV-1 processes required for tRNA upregulation.** Human fibroblast cells were mock-infected or infected with ΔICP4 (n12), ΔICP27 (5dl1.2), ΔICP0 (n212), ΔICP22 (n199), ΔICP0/4/22/27/47 (d109), ΔUL30 (hp66), or wild-type HSV-1. RNA was isolated at 12 hpi, and **A**, **B** Northern blots, or **D–F** Ribo-depleted Total RNA-Seq was used to assess transcript abundance. **A**, **B** Representative Northern blot images. Northern blots were performed for a minimum of two biological replicates, and quantification is in Supplementary Fig. 5. **C** Summary of phenotypic differences between HSV-1 mutants used. **D** Heatmaps for viral transcripts clustered by kinetic class with yellow as immediate early, green as early, blue as leaky late (L1), purple as true late (L2), and black as latency (LAT). Transcripts highlighted with asterisks are upregulated (log₂ fold change > 0) in all mutants relative to ΔUL30 infection. **E**, **F** All RNA-Seq reads mapping to the host or viral genome assembly plotted as abundance (mapped reads per million ERCC spike-in reads, MR/MSI) or as the ratio of normalized viral transcripts: normalized host transcripts (Viral:Host Ratio).

viral transcript accumulation, thus mutants defective for these proteins had significantly reduced viral transcript levels, ~7-fold compared to wildtype (Fig. 3E–F). Supporting our hypothesis, ΔUL30 was the most defective for total viral transcription, ~20-fold compared to wildtype (Fig. 3E–F). These data taken together conclude that tRNA upregulation requires nuclear viral entry, but not synthesis of specific viral transcripts, nascent viral genomes, or viral progeny. These findings support a link between the amount, but not identity, of viral transcripts and tRNA upregulation.

**Changes to Pol III GTF binding after HSV-1 infection.** Next, we tested if infection altered recruitment of the Pol III transcription machinery to tRNA loci. We also assessed recruitment of the catalytic subunit of Pol II, POLR2A. We performed ChIP-Seq in MRC5 cells mock-infected or infected with HSV-1 for 2, 4, or 6 h. Of the transcription machinery tested—POLR3A, BRF1, GTF3C5, TBP, POLR2A—we found that POLR2A had increased binding, by ~2-fold, to tRNA loci (Fig. 4, Supplementary Fig. 7). Binding of TBP to tRNA loci also showed a moderate increase, but not as strong as POLR2A. Concurrent with prior findings, POLR2A binding to mRNA genes decreased drastically during infection, a phenomenon that promotes host shut off[4–7] (Fig. 4A). tRNA loci with increased POLR2A recruitment were found in accessible regions of the genome (Fig. 4B). These same tRNA loci were upregulated after infection in our DM-tRNA-Seq dataset (Fig. 4B–C). We did not observe increased recruitment of the catalytic subunit of Pol III, POLR3A, despite our earlier finding that tRNA loci had increased transcriptional output (Fig. 2D–E).

To determine whether transcriptional host shut off and tRNA upregulation may be linked phenomena, we performed ChIP-Seq for POLR2A on human fibroblasts after infection with ΔICP0/4/22/27/47, ΔICP4, ΔICP27, ΔICP22, and wildtype HSV-1 for 6 h (Supplementary Fig. 8). We observed increased recruitment of POLR2A to tRNA loci during ΔICP4, ΔICP27, ΔICP22, and wildtype HSV-1 infection. Of the mutants tested, ΔICP0/4/22/27/47 infection had the lowest level of POLR2A recruitment to tRNA loci (Supplementary Fig. 8). Host shut off is dependent on the presence of ICP4, and scales with viral genome copy number[6,7]. Consistent with this, we only observed depletion of Pol II from host promoters after infection with ΔICP27, ΔICP22, and wildtype HSV-1 infection (Supplementary Fig. 8). Minimal Pol II recruitment to tRNA loci after ΔICP0/4/22/27/47 infection agrees with our earlier findings that this mutant does not cause tRNA upregulation (Figs. 1 and 3).

To assess how HSV-1 promotes Pol II recruitment at tRNA loci we investigated changes in transcription factor expression and availability. We performed polyA-selected and RiboMinus RNA-Seq on human fibroblasts mock-infected or infected with HSV-1 (Supplementary Fig. 9). PolyA-selected RNA-Seq quantifies the amount of a given mRNA in the total population. However due to comprehensive changes in the population of mRNA transcripts during HSV-1 infection, RiboMinus RNA-Seq allows for a more quantitative assessment. In line with the global reduction of host mRNA species following HSV-1 infection (Supplementary Fig. 9A–B), by 12 hpi transcripts for most of the Pol II and III machinery had decreased between 2 to 32-fold (Supplementary Fig. 9C–D). The one exception was POLR2A, for which we observed a slight increase in transcript abundance by polyA-selected RNA-Seq—but not RiboMinus RNA-Seq—peaking around 4 hpi (Supplementary Fig. 9C). We next assessed whether these transcriptional changes resulted in altered protein expression (Supplementary Fig. 10). While POLR2A transcript abundance increased around 4 hpi, its protein expression levels decreased ~4-fold by 12 hpi. Again, we observed a marked decrease in protein expression levels for all transcription machinery assessed, with the exception of POU2F1 and SP1. These cellular activators are of particular note because they promote Pol III Type III transcription and are also critical for VP16-mediated enhancement of viral immediate early transcription[27]. Since these activators are not required for Pol III type II transcription, we find it unlikely they contribute to tRNA upregulation. These results rule out a model in which increased Pol III machinery abundance promotes increased tRNA transcriptional output.

Another mechanism by which HSV-1 alters the host environment is by remodeling the nucleus into sub-domains. HSV-1 assembles within replication compartments enriched for utilized

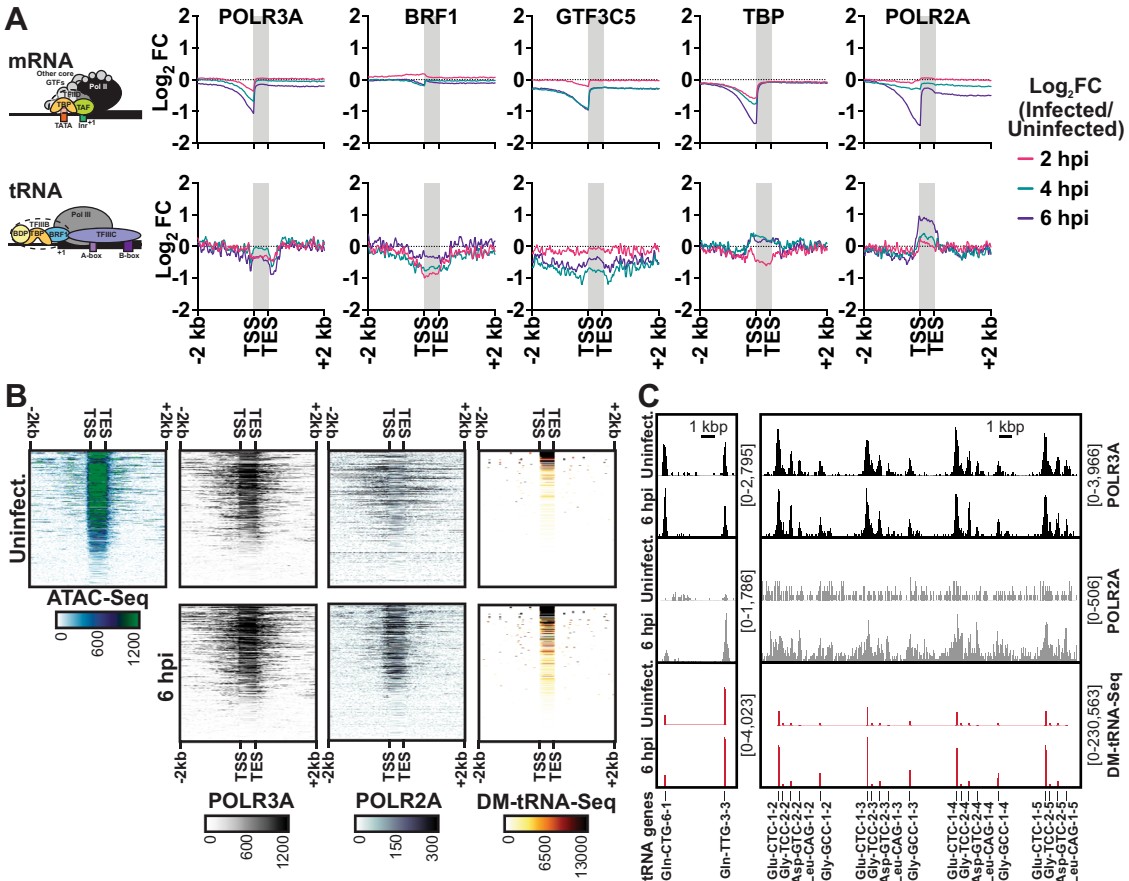

**Fig. 4 Changes to Pol III GTF binding after HSV-1 infection. A–C** Human fibroblasts were mock-infected or infected with HSV-1 and ChIP-Seq, ATAC-Seq, or DM-tRNA-Seq was performed. ChIP-Seq data is from biological duplicate experiments and normalized for sequencing depth and cellular genome sampling. ATAC-Seq data is from biological duplicate experiments and normalized for sequencing depth, excluding MT reads. DM-tRNA-Seq data was normalized to an internal spike-in control. **A** The average log$_2$ fold change for infected over the matched uninfected dataset from 2 kb upstream of TSS to 2 kb downstream of TES for all mRNA or tRNA loci. Individual replicates which make up the average are plotted in Supplementary Fig. 7. **B** Heatmaps for normalized sequencing data mapped from 2 kb upstream of TSS to 2 kb downstream of TES for tRNA loci. **C** Traces of representative tRNA genes, y-axes minimum and maximum are shown in brackets.

host factors, such as transcription and replication machinery, while excluding negative host factors, most notably histones[28]. We imaged Vero cells that were uninfected (Supplementary Fig. 11A), infected pre-replication (Supplementary Fig. 11B), and infected post-replication (Supplementary Fig. 11C–D). We used the nucleoside analog, EdC, to specifically label viral genomes and then stained for various components of the transcriptional machinery including: POLR2A, POLR3A, POLR3G, BRF1, and GTF3C5. We observed a strong reorganization to viral replication compartments for all host factors tested. Of note, only POLR2A colocalized with viral input genomes (Supplementary Fig. 11B). All other components colocalized with the viral genome only after formation of viral replication compartments and multiple rounds of viral genome replication had occurred (Supplementary Fig. 11D). Based on these results, it is unlikely that HSV-1 promotes Pol II recruitment to tRNA loci by enhancing the local concentration gradient. Additionally, most factors reorganized to viral replication compartments (Supplementary Fig. 11), suggesting decreased recruitment to the host genome—an observation in line with our ChIP-Seq results showing global decrease of POLR2A and TBP from host mRNA promoters and slight decrease of POL3A, BRF1, and GTF3C5 from tRNA loci (Fig. 4A). These results mirror the global environment of host transcriptional repression present in HSV-1 infection, wherein tRNA appear to be the rare outlier.

**Recruitment of Pol III to the HSV-1 genome**. Classically all HSV-1 transcripts are defined as Pol II-dependent[29]. However, there are examples of other DNA viruses that encode Pol III transcripts; the VAI and VAII RNAs of adenovirus[8], EBER1 and EBER2 of Epstein Barr Virus[9], and the TMERs of MHV68[10]. We examined our ChIP-Seq data for binding of the Pol III machinery to the HSV-1 genome. We observed strong, distinct recruitment of the catalytic subunit of Pol III, POLR3A, to the viral genome (Fig. 5).

Before viral replication, POLR3A binding to the HSV-1 genome clustered most closely with POLR2A (Fig. 5A). POLR3A in all conditions had the highest number of intersecting peaks with POLR2A. Based on the degree of intersection (Jaccard statistic), POLR3A also highly overlapped with GTF3A and at later times TBP (Fig. 5B). After replication, POLR2A more closely clustering with TBP and the viral transcription factor, ICP4 (Fig. 5A). This was because in the presence of ICP4 and/or after viral replication the number of POLR2A peaks increased, but the number of POLR3A peaks did not (Fig. 5B–C). We observed few instances (between 3 to 5), in which POLR3A bound independent of POLR2A (Fig. 5B–D). A notable example occured within the latency-associated transcript (LAT) where the POLR3A peak did not overlap with a characterized Pol II-dependent promoter (Fig. 5D). This POLR3A peak was one of the most prominent in pre-replication conditions (<2.5 hpi) and nearly absent post-replication (Supplementary Fig. 12 and 13).

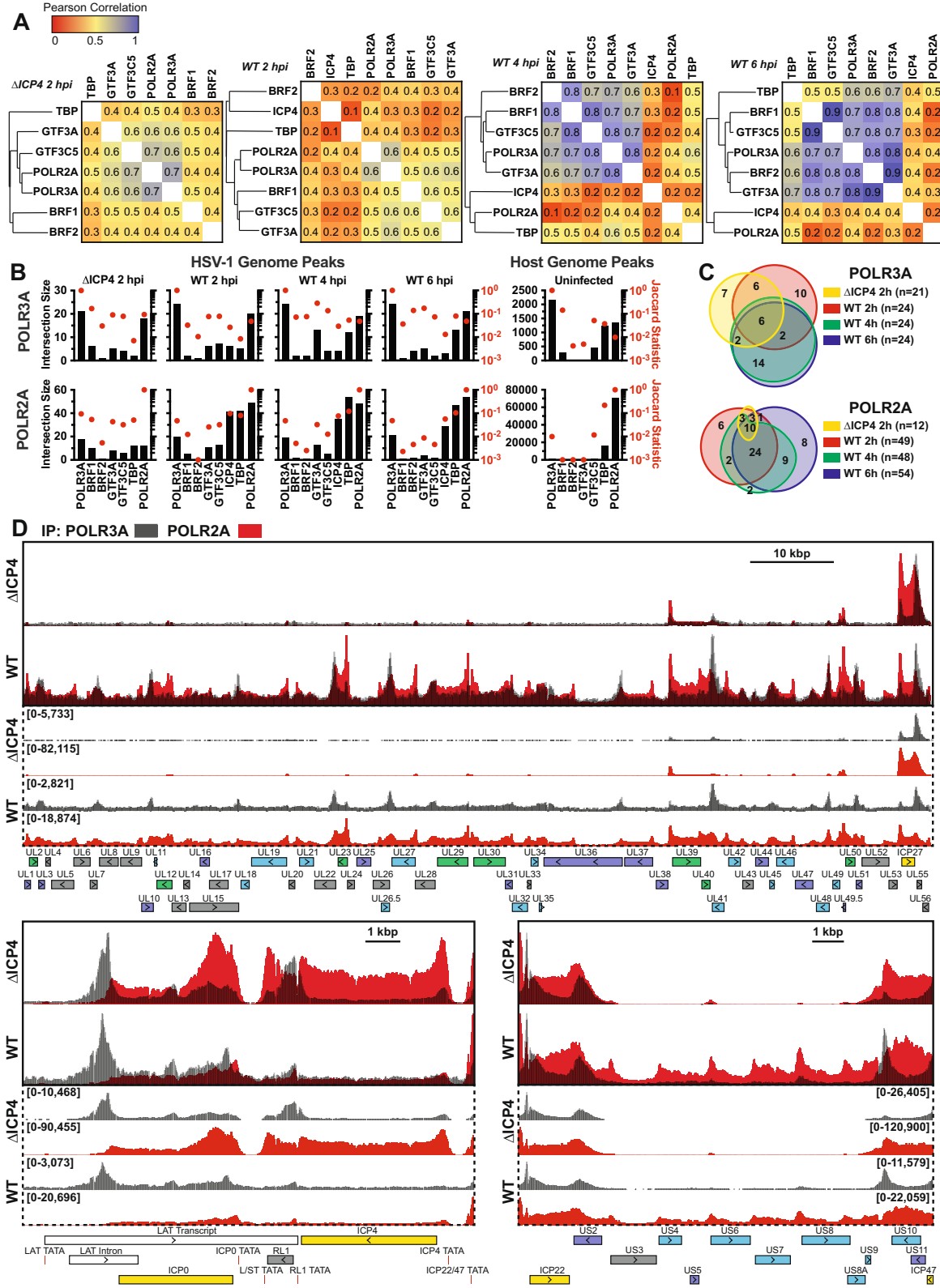

We characterized Pol III transcription factor data for the host genome to compare and contrast binding phenotypes. As expected we observed strong intersection of GTF3A, BRF1, GTF3C5, and BRF2 binding at characterized Pol III-dependent Type I, II, or III genes (Supplementary Fig. 14). Pol III has previously been characterized to bind strong host Pol II-dependent promoters[13]. We observed this in our own data, and found that POLR3A binding clustered closely with POLR2A and TBP on host Pol II-dependent protein-coding genes (Supplementary Fig. 12A–B). POLR3A clustering on host Pol II promoters resembled POLR3A clustering on the viral genome pre-replication (Supplementary Fig. 14B and Fig. 5A).

**Fig. 5 Recruitment of Pol III to the HSV-1 genome.** Human fibroblasts were infected with ΔICP4 (n12) or wildtype HSV-1. ChIP-Seq was performed for POLR3A, POLR2A, TBP, GTF3A, GTF3C5, BRF1, BRF2, and ICP4. Data is the average of biological duplicates and normalized for sequencing depth and viral genome copy number. **A** Pearson correlation analysis of binding profiles for the viral genome, these were calculated by binning (10 bp) normalized traces. **B** Intersection analysis for POLR3A and POLR2A relative to other factors tested for viral and host peaks. Intersection size is the number of instances where peaks overlap by at least 1 nucleotide. Jaccard statistic is the degree of overlap between all peak regions, with the maximum value being 1. **C** Number of overlapping peaks on the viral genome for POLR3A and POLR2A between conditions tested. **D** Traces of POLR2A and POLR3A binding to the unique long (UL), joint, and unique short (US) regions of the HSV-1 genome. Y-axes maximum and minimum values are listed within brackets. Viral CDS are listed below, with colors indicating kinetic transcriptional class: immediate early (yellow), early (green), leaky late (blue), true late (purple), unclassified (gray). We have also annotated additional genomic features such as TATA boxes (red) and ncRNA (white).

**Unique GTF context of viral Pol III binding**. A closer examination of the Pol II-independent POLR3A binding event within the latency locus revealed strong BRF2 recruitment (Fig. 6A). BRF2 binds a very small repertoire of host promoters (around a dozen targets[11]), an observation consistent with our own analysis of host binding (Supplementary Fig. 14, Fig. 6B). Classically BRF2 aids in transcription of Pol III-dependent Type III transcripts, and forms a TFIIIB complex containing BDP1 and TBP. At 2 hpi, BRF2 bound robustly within the LAT intron, coincident with POLR3A and GTF3C1, but not TBP, GTF3C2, GTF3C3, GTF3C4, GTF3C5, or GTF3C6 (Fig. 6, Supplementary Fig. 12). This binding context was distinct from what we observed for BRF2 binding on the host genome—in which BRF2 is recruited coincident with TBP and to a lesser extent GTF3A and GTF3C5 (Fig. 6B, Supplementary Fig. 14).

We also assessed recruitment of all six components of the TFIIIC complex. On the host genome, GTF3C1-6 are recruited together at Pol III-dependent Type I and II promoters (Fig. 6C, Supplementary Fig. 15). We will note that our immunoprecipitation of GTF3C1-3 was not as robust as GTF3C4-6. This resulted in a smaller number of peaks called on the host genome, however of those peaks less than 1% of all GTF3C1-3 peaks bound independent of GTF3C4-6 (Fig. 6C). GTF3C1-6 binding on the viral genome did not overlap to the same extent (Fig. 6C). Recruitment of GTF3C2 and GTF3C3 was largely absent, and signal mimicked matched input ChIP data (Fig. 6D). GTF3C5 and GTF3C6 were recruited and largely coincident with POLR3A and POLR2A binding events (Fig. 6D). The most interesting was GTF3C1, who's HSV-1 genome binding differed from all other TFIIIC subunits (Fig. 6C–D). Overall the recruitment of BRF2 and TFIIIC to the HSV-1 genome differs from promoter contexts observed and characterized for the host genome.

## Discussion

Herein we characterized how HSV-1 affects Pol III mediated transcription (Fig. 7), a pathway commonly dysregulated by cancer and pathogens. In prior work, HSV-1 has been shown to induce the Pol III type II transcript, Alu[21,22]. We found that HSV-1 targets and upregulates tRNA, another group of Pol III type II transcripts. Within 12 h of HSV-1 infection, total- and nascent-tRNA abundance increased 2- and 10-fold, respectively. We identified 15 mature- and 66 pre-tRNA species upregulated within 12 h of HSV-1 infection. HSV-1 mediated tRNA upregulation is at odds with the general environment of host transcriptional shut off mediated by the virus, including downregulation of most protein-coding mRNA and Pol III type III transcripts. We propose that HSV-mediated tRNA upregulation is required for robust productive replication, wherein a single infected cell can synthesize up to 30,000 viral progeny within a 12-hour window. During this time the virus must produce ~90 viral proteins at exponential rates to facilitate genome replication and virion assembly. Alternatively, increased tRNA expression may be critical in infection scenarios where host shut off is less

pronounced and there is more competition for translation resources.

Using a combination of approaches we explored the mechanism by which HSV-1 causes tRNA upregulation. We observed a 10-fold increase in nascent-tRNA levels, but not increased recruitment of Pol III to tRNA. Our results support a model in which tRNA upregulation is caused by a simultaneous increase in Pol III initiation and elongation rates[30]. Considering the short half-life of pre-tRNAs, decreased turnover may also contribute to increased nascent abundance. To determine the mechanism by which HSV-1 induces tRNA transcription, we tested various hypotheses including altered Pol III transcription factor expression, recruitment, or localized concentration. HSV-1 productive infection resulted in a decrease for all three. Decreased Pol III machinery expression suggests that the mechanism by which HSV induces tRNA synthesis differs from other viruses. Namely adenovirus[15,16], SV40[17], and Epstein Barr Virus[18] which increase the abundance of Pol III transcription factors by viral regulatory proteins E1A, E1B, T-antigen, and EBNA1. We then turned to another possible option, namely crosstalk between the Pol II and III transcription machinery. Recent work has increasingly demonstrated a dependence and interplay between the Pol II and III machinery[11–14,31]. In a global environment of Pol II depletion from host genes, we were surprised by a 2-fold increase in Pol II recruitment to tRNA loci. Recruitment of Pol II to tRNA genes only occurred upon infection of HSV-1 mutants that also had increased tRNA abundance. Additionally, recruitment of Pol II to tRNA genes occurred independent of host transcriptional shut down. The question now becomes, how is Pol II enhancing tRNA transcription rates? We propose three potential options: i. Pol II alters the chromatic environment at bound tRNA, ii. enhances the rate of Pol III termination or re-initiation, or iii. functions itself to transcribe tRNA.

Infection with a panel of HSV-1 mutants allowed us to characterize viral life events critical for tRNA upregulation. We found that nuclear viral entry was required, but synthesis of specific viral transcripts, genomes, or progeny was not required. Furthermore host shut off was not necessary for tRNA upregulation, suggesting a different mechanism of induction than MHV68 perturbation of the tRNA landscape[19]. Our results define a narrow window of viral processes which may be critical for tRNA upregulation, namely entry of the viral genome and synthesis of viral transcripts/proteins—with no dependence on exactly which genes are expressed. We posit that changes in the tRNA pool are either induced by a shared feature of viral transcripts/proteins, or in response to the homeostatic toll rapid expression of viral genes tasks on the host cell. We propose a few possibilities, first HSV-1 transcripts have an average GC content of ~68%, propensity to form G-quadruplexes, and high density of complementary transcripts. Second, HSV-1 proteins are composed of a high density of intrinsically disordered domains, and due to the rapid rate of synthesis are prone to misfolding and aggregation—traits that robustly trigger an unfolded protein response. tRNA abundance profiles are known to change drastically in response to various stimuli including oxidative stress, osmotic stress, temperature

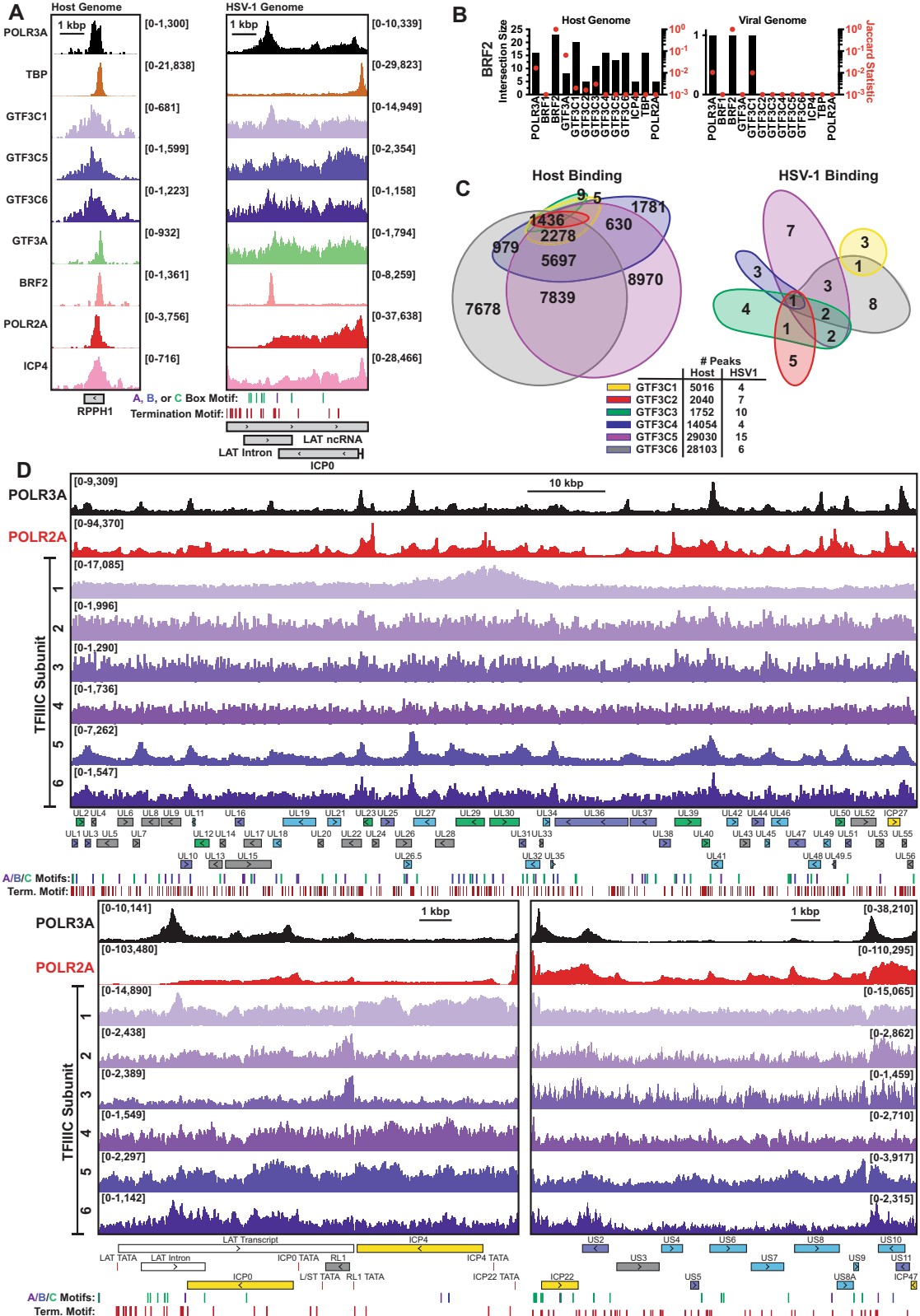

stress, and diauxic shift[32]. Which leads to our final hypothesis, in which tRNA may be induced by virus perturbation of host homeostasis including altered levels of metabolites including nucleotides and amino acids, inducing an unfolded protein response, and dysregulating the DNA damage response[33]. Any one of these cellular feedback loops could be hijacked by HSV-1 to increase the pool of tRNA immediately available for viral use.

Thus far, we have focused on how the virus alters host Pol III transcription. However, various viruses including adenovirus, Epstein Barr Virus, and murine herpesvirus-68 possess Pol III-dependent viral transcripts[8–10]. These RNAs have functions that combat the innate antiviral response, contribute to latency, and play a role in transformation. Our data suggests that Pol III may play roles in HSV-1 transcription. We observed Pol III binding to

**Fig. 6 Unique GTF context of viral Pol III binding.** Human fibroblasts were infected with HSV-1 for 2 h. ChIP-Seq was performed for POLR3A, TBP, GTF3C1, GTF3C2, GTF3C3, GTF3C4, GTF3C5, GTF3C6, GTF3A, BRF2, POLR2A, and ICP4. Data is the average of biological duplicates and normalized for sequencing depth (scaling for host and viral traces are identical). **A** ChIP-Seq data mapping to the viral or host genome at regions of strong BRF2 recruitment. Sequences matching Pol III A-, B-, or C-box motifs and termination signals (TTTT) are annotated underneath. **B** Peak-based intersection analysis for BRF2 binding relative to other factors tested. Intersection size is the number of instances where peaks overlap by at least 1 nucleotide. Jaccard statistic is the degree of overlap between all peak regions, with the maximum value being 1. **C** Number of overlapping TFIIIC complex peaks on the viral and host genome. For host binding, only the top six highest intersections as well as non-overlapping regions have values annotated. **D** Traces of POLR3A, POLR2A, and TFIIIC binding to the unique long (UL), joint, and unique short (US) regions of the HSV-1 genome. Y-axes maximum and minimum values are listed within brackets. Viral CDS are listed below, with colors indicating kinetic transcriptional class: immediate early (yellow), early (green), leaky late (blue), true late (purple), unclassified (gray). Viral ncRNA, namely LAT, are white. Sequences matching Pol III A-, B-, or C-box motifs and termination signals (TTTT) are annotated underneath.

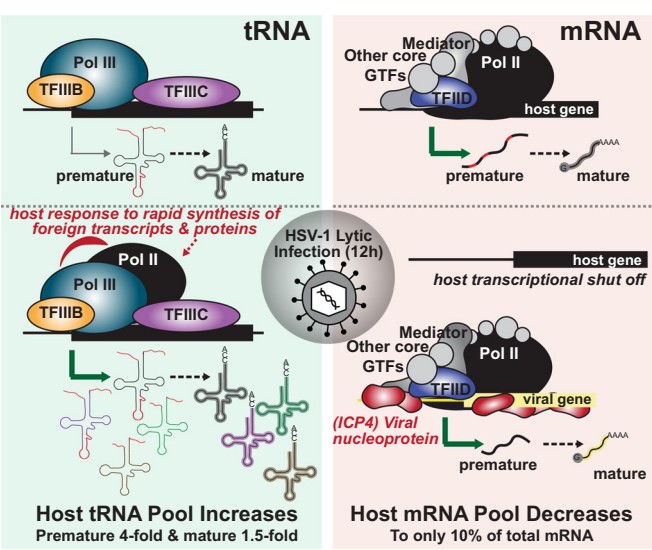

**Fig. 7 Model of HSV-1 Pol II and III Modulation during lytic infection.** During lytic infection, HSV-1 alters host Pol II recruitment and induces depletion from host mRNA in favor of host tRNA and viral mRNA. With 12 h, these changes result in a 2-fold increase in total abundance of tRNA and a 10-fold increase in nascent levels of tRNA. In contrast, host mRNA decreases to only 10% of the total present in the cell. These phenomena are mediated distinctly. Depletion of Pol II from mRNA promoters scales with viral genome replication and ICP4. Recruitment of Pol II to tRNA promoters, and their subsequent upregulation, is caused by entry of the viral genome and synthesis of viral transcripts—with no dependence on exactly which transcripts are made.

the viral genome in two conformations: (i) coincident with RNA Pol II, or (ii) coincident with the canonical Pol III type III transcription factor, BRF2. The latter binding event occurred in a region of the genome where latency-derived transcripts originate, suggesting a cell-type specificity and potential role in viral persistence and reactivation. The L/S joint region of the viral genome is rich with ncRNA, including LAT[34], L/STs[35], and microRNAs[36]. Pol II promoters have been associated with the genesis of the LAT and L/STs[37,38], and we have also shown that these promoters function in a reconstituted Pol II in vitro transcription systems[39,40]. However, these studies do not preclude the possibility that there are alternative mechanisms to transcribe these RNAs, the microRNAs, or other noncoding RNAs yet to be discovered. Herein we provide evidence suggesting a previously unrecognized role for Pol III in HSV-1 gene expression.

## Methods

**Cells and viruses.** Vero (African green monkey kidney, CCL-81), U2OS (human osteosarcoma, HTB-96), and MRC5 (human fetal lung, CCL-171) cells were obtained from and propagated as recommended by ATCC.

### Table 1 HSV-1 viruses used in study.

| Virus | Mutant genotype | Background strain |
|---|---|---|
| **n199**[63] | ICP22 nonsense | KOS |
| **n212**[64] | ICP0 nonsense | KOS |
| **d99**[65] | ICP0 deletion | KOS |
| **n12**[24] | ICP4 nonsense | KOS |
| **d120**[24] | ICP4 deletion | KOS |
| **5dl1.2**[66] | ICP27 deletion | KOS |
| **d92**[42] | ICP4/27 deletion | KOS |
| **d109**[65] | ICP0/4/22/27/47 deletion | KOS |
| **R3616**[67] | RL1 deletion | F |
| **F-ΔICP47**[68] | ICP47 deletion | F |
| **F-ΔICP47-R**[68] | ICP47 deletion revertant | F |
| **hp66**[44] | UL30 deletion | KOS |
| **KOS**[69] | Wildtype | N/A |
| **F-Strain** | Wildtype | N/A |

Table 1 contains a detailed list of all viruses used in this study. n199, R3616, F-ΔICP47, and KOS were prepared and titered in Vero cells. Other virus stocks were prepared and titered in the following Vero-based complementing cell lines: E5 (ICP4 +, n12, d120-complementing[41]); E11 (ICP4/ICP27+, 5dl1.2, d92-complementing[42]); F06 (ICP4/ICP27/ICP0 +, d109-complementing[43]); U2OS (n212, d99-complementing), POLB3 (hp66-complementing[44]). We thank Don Coen (hp66, POLB3), David A. Leib (R3616), and Anthony St. Leger (F-ΔICP47, F-ΔICP47-R) for their kind gift of viruses and/or cells.

**Antibodies.** A list of all antibodies and the amount used per technique is located in Supplementary Table 1.

**Oligos.** A list of all oligos used for either qPCR or Northern Blot is located in Supplementary Table 2.

**Viral infection.** Confluent cell monolayers were infected with 10 PFU per cell. Virus was adsorbed in tricine-buffered saline (TBS) for 1 h at room temperature. Viral inoculum was removed, and cells were washed quickly with TBS before adding 2% FBS media. 0 h time point was considered after adsorption of infected monolayers when cells were place at 37 °C to incubate.

**qPCR quantification of mRNA.** Cell monolayers were collected by aspirating supernatant and washing twice with TBS. Cells were scraped into 1 mL TBS and pelleted, supernatant was discarded. RNA was isolated using the RNAqueous Micro Kit (ThermoFisher cat. no. AM1931). cDNA was generated from 500 ng total RNA, as quantified using the Agilent RNA 6000 Nano Kit. RNA was reverse transcribed with 20 units Riboguard RNase inhibitor, 2.5 uM Random decamer primer (Invitrogen cat no. 5722 G), 100 4 units MMLV-HP reverse transcriptase, 10 mM dithiothreitol, 2.5 mM dNTPs, and 1x reaction buffer (Epicentre cat no. RT80125K and RG90910K). RNA and random decamer primer were first incubated at 85 °C for 3 min and then incubated on ice for 2 min during which remaining reaction components were added. The entire reaction was incubated at 65 °C for 2 min and then 37 °C for 1 h. To heat inactivate components the cDNA was incubated at 85 °C for 5 min. Standard curves were generated using purified HSV-1 (KOS) or human genome stocks.

**Western blot.** At the indicated times postinfection, proteins were isolated from cells using Laemmli SDS sample buffer and Western blotting was carried out using the primary antibodies listed in Table 1. Blot were probed with secondary

antibodies—IRDye goat anti rabbit or goat anti-mouse 680/800—at a 1:15000 dilution. The intensity of gel bands was quantified using the Odyssey CLx system machine and Image Studio program. Band intensities were normalized to loading controls including GAPDH, alpha-tubulin (TUB4A), or Vinculin (VCL) detected from the same sample.

**Northern blot**. Total RNA was isolated from cells using TRIzol (Invitrogen), following manufacturer's instructions. RNA was quantified using Agilent RNA 6000 Nano kit. 5–20 μg of RNA was loaded into 10% TBE-Urea Polyacrylamide Gel, transferred to Hybond-N + (Sigma) membranes using the Owl™ HEP Series Semidry Electroblotting Systems (ThermoFisher). Blots were pre-hybridized in ExpressHyb buffer (Takara Bio) at 37 °C for 1 h. If using radiolabeled probes specific to tRNAs, 50 nM unfolder oligo (see Supplementary Table 2) was included in the pre-hybridization step, as described in[45]. Probes were generated by end-labeling oligos listed in Supplementary Table 2 using T4 PNK and [g-32P]- ATP. Probes were hybridized to the blots in ExpressHyb buffer at 37 °C overnight. If using radiolabeled probes specific to tRNAs, 25 nM unfolder oligo was included in the hybridization step. After hybridization, blots were washed four time in 2× SSC [0.3 M NaCl, 0.03 M Trisodium Citrate pH 7.0], 0.05% SDS and 2 times in 0.1× SSC, 0.1% SDS. Blots were exposed and quantified using the Typhoon Biomolecular Imagine (Amersham). Blots were stripped in 1% SDS, 0.1× SSC, 40 mM Tris pH 8 for four 10 min washes at 70 °C. Northern blot quantifications are plotted as "Relative Level", which is the band of interest relative to the paired uninfected sample run on every blot. 5.8 S rRNA was quantified for each blot and plotted— serving as a loading control. Images are representative of at least two biological replicates, with the replicates quantified and plotted in the main paper or supplemental materials.

**Immunofluorescence**. $1.7 \times 10^5$ Vero cells were infected as described above. Coverslips were fixed at indicated time point with 3.7% paraformaldehyde. EdC labeling of viral replication compartments, click chemistry, and immuno-fluorescence were conducted as previously described[28,46]. Cellular DNA was stained with 1:2000 Hoescht, and immunofluorescence was carried out using antibodies listed in Table 1 and 594-conjugated secondary antibodies (Santa Cruz, 1:500). Images are taken using an Olympus Fluoview FV1000 confocal microscope. Images are representative of biological duplicate experiments.

**DM-tRNA-Seq**. Protocol was modified from[19,25], with the following modifications. $7 \times 10^7$ MRC5 cells were infected as described above and total RNA was isolated from cells using TRIzol (Invitrogen), following manufacturer's instructions. Total RNA extracted from two biological replicates was spiked with in vitro transcribed *E. coli* tRNA-Lys, *E. coli* tRNA-Tyr, and *S. cerevisiae* tRNA-Phe transcripts at 0.01 pmol IVT tRNAs per μg total RNA. RNA was deacylated in 0.1 M Tris-HCl, pH 9 at 37 °C for 45 min, ethanol purified, and then dephosphorylated with PNK. Deacylated and dephosphorylated RNAs were purified with a mirVANA small RNA purification kit (Ambion). RNAs were demethylated in 300 mM NaCl, 50 mM MES pH 5, 2 mM MgCl2, 50 μM ferrous ammonium sulfate, 300 μM 2-ketoglutarate, 2 mM L-ascorbic acid, 50 μg/ml BSA, 1U/μl SUPERasin, 2X molar ratio of wt AlkB, and 4X molar ratio of D135S AlkB for 2 h at room temperature. Ni-NTA cation exchange purified His-tagged wild-type and D135S AlkB. Reaction was quenched with 5 mM EDTA and purified with Trizol LS reagent. Two TGIRT reactions from each biological replicate was performed, these are considered technical replicates. In total each sample condition is an average of 4 data points, consisting of two biological replicates containing technical duplicates. 100 ng demethylated small RNAs was used for library prep with a TGIRT Improved Modular Template- Switching RNA-seq Kit (InGex) following the manufacturer's instructions. PCR amplification was performed with Phusion polymerase (Thermo Fisher) with Illumina multiplex and barcoded primers. Libraries were quantified using the Agilent DNA 7500 Kit, and samples were mixed together at equimolar concentration. Samples were size selected for 150-250 bp fragments using the Pippin system. Illumina NextSeq 550 platform was used to generate 75 bp PE reads and carried out at the Tufts University Core Facility.

**4SU-Sequencing**. $2 \times 10^6$ MRC5 cells were infected with wild-type HSV-1 (KOS) as described above. At indicated time (hpi) 500 uM 4SU (Sigma) was added to cell culture medium. 15 min post-4SU addition, cells were collected and RNA extracted using miRNeasy kit (Qiagen). 50-100 ug total RNA was biotinylated in 10 mM Tris pH 7.4, 1 mM EDTA, 0.2 mg/mL EZ-link Biotin-HPDP (ThermoFisher). Unbound biotin was removed by performing a chloroform:isoamyl alcohol extraction using MaXtract High Density tubes (Qiagen). RNA was isopropanol precipitated and resuspended in water. Biotinylated RNA was bound 1:1 to Dynabeads My One Streptavidin T1 equilibrated in 10 mM Tris pH 7.5, 1 mM EDTA, 2 M NaCl. Bound beads were washed three times with 5 mM Tris pH 7.5, 1 mM EDTA, 1 M NaCl. 4SU-RNA was eluted with 100 mM DTT and isolated using the RNeasy MinElute Cleanup Kit (Qiagen). RNA-Seq libraries were generated using NEBNext Ultra Directional RNA Library Prep Kit for Illumina. Libraries were quantified using the Agilent DNA 7500 Kit, and samples were mixed together at equimolar concentration. One biological replicate was sequenced using the Illumina HiSeq 2500

platform to generate 50 bp SE reads and carried out at the Tufts University Core Facility.

**PolyA-selected RNA-Seq**. Total RNA was harvested using the Ambion RNAqueous-4PCR kit and quantified using the Agilent RNA 6000 Nano kit. RNA-Seq libraries were generated from 2 μg RNA using NEBNext Poly(A) mRNA Magnetic Isolation Module and NEBNext Ultra Directional RNA Library Prep Kit for Illumina (NEB #E7490 and #E7420). Libraries were quantified using the Agilent DNA 7500 Kit, and samples were mixed together at equimolar concentration. Two biological replicates were sequenced using the Illumina HiSeq 2500 platform to generate 50 bp SE reads and carried out at the Tufts University Core Facility.

**Ribominus total RNA-Seq**. Total RNA was isolated from cells using TRIzol (Invitrogen), following manufacturer's instructions. RNA was quantified using Agilent RNA 6000 Nano kit. ERCC spike-in controls (ThermoFisher) were added to 500 ng of total RNA and ribominus selection was performed using the NEBNext® rRNA Depletion Kit. RNA-Seq libraries were generated using the NEBNext Ultra Directional RNA Library Prep Kit for Illumina. Libraries were quantified using the Agilent DNA 7500 Kit, and samples were mixed together at equimolar concentration. Two biological replicates for mock- and wildtype HSV-infected samples and one biological replicate for mutant HSV infected samples was sequenced using the Illumina NextSeq550 platform to generate 75 bp PE reads and carried out at the Tufts University Core Facility.

**ChIP-Sequencing**. ChIP-Seq was performed on mock-infected or infected MRC5 cells as described previously[7], with antibodies listed in Table 1. Input samples were taken after chromatin was sheared by sonication and prior to antibody-based immunoprecipitation. Libraries were quantified using the Agilent DNA 7500 Kit, and samples were mixed together at equimolar concentration. For each IP and paired input sample at least two biological replicates were sequenced using the Illumina HiSeq 2500 platform to generate 50 bp SE reads and carried out at the Tufts University Core Facility.

**ATAC-Sequencing**. We adapted the protocol from Buenrostro et al. (2013), and previously published the data see SRA# PRJNA553559[7].

**Bioinformatic analysis**. Data was uploaded to the Galaxy web platform, and we used the public server at usegalaxy.org to analyze the data[47].

**DM-tRNA-Seq**. We quantified mature and premature tRNA DM-tRNA-Seq datasets similar to the analysis described in[19]. Reads were trimmed using Cutadapt to remove adapters, artificially appended G's generated as an artifact of the NextSeq platform, and sequences below the quality cutoff of 30[48]. Reads were filtered to only retain species with a minimum length of 50 and maximum length of 200, reducing fragmented or partial reads which may introduce error during quantification. All mapping steps were performed using Bowtie2 with the following settings: -I: 50, -X: 300, -phred33, -end-to-end, -N: 0, -L: 20, -i: S,1,0.5, -n-ceil: L,0,0.15, -k: 2, -D: 20, -re-seed: 3. To discriminate between pre- and mature-tRNA species we first mapped data to an assembly of mature-tRNA sequences wherein 5' and 3' leader sequences are absent, introns are spliced, and 3' CCA tails are present. Mature and premature-tRNA assemblies for hg38 were generated using the tRAX pipeline[49]. Mitochondrial and nuclear tRNA loci were based on GtRNAdb[50,51] and tRNAscan-SE[52] for hg38. Unaligned reads were then mapped to a modified host (hg38) genome in which tRNA genes were masked and instead appended as an additional chromosome containing pre-tRNA sequences with introns, and 5' and 3' leader sequences. We assessed whether the length of reads mapping to mature and premature-tRNA assemblies matched expectation using CollectInsertSizeMetrics. The median size for mature-tRNA mapped reads was 70-75 nt. Reads mapped to premature-tRNA has a multi-modal distribution peaking around 76, 87, and 100 nt. These sizes gave increased confidence regarding accurate discrimination of pre- and mature-tRNA. Transcript quantification was performed individually on mature- and premature-tRNA BAMs using salmon quant in alignment mode[53]. Raw counts were normalized relative per million in vitro transcribed tRNA spike-in controls and per kilobase pair. Assessment of differentially expressed tRNA was performed using normalized counts matrices in EdgeR with a *p*-value adjusted (Benjamini-Hochberg, TMM normalization) threshold of 0.05 to achieve significance[54]. As our pipeline maps sequentially to mature- and then premature-tRNA, the amount of pre-tRNA reads quantified will be an underestimate. Additionally due to high sequence overlap between tRNA genes for the same isodecoder, unless the premature reads contain locus specific flanking regions or introns these reads will be split between genes encoding the same isodecoder.

**4SU-Seq**. Similar to our DM-tRNA-Seq mapping strategy we used Bowtie2 with the following settings: -I: 50, -X: 300, -phred33, -end-to-end, -N: 0, -L: 20, -i: S,1,0.5, -n-ceil: L,0,0.15, -k: 2, -D: 20, -re-seed: 3. Reads were first mapped to an assembly of mature-tRNA sequences and then unaligned reads were mapped to a modified host genome in which tRNA genes were masked and instead appended as an additional chromosome. Transcript quantification was performed individually

on mature- and premature-tRNA BAMs using salmon quant in alignment mode[53]. We did not deplete rRNA species prior to sequencing, and rRNA comprised ~40% of total reads. As rRNA levels do not shift during HSV-1 infection, we quantified 4SU-Seq reads as mapped reads per billion rRNA reads per kbp.

**Transcriptomic analysis: ribominus total RNA-Seq, PolyA-selected RNA-Seq.** To assess mRNA levels, Ribominus Total or PolyA-selected RNA-Seq data was aligned sequentially using HISAT2 to the human genome (hg38) and HSV-1 strain KOS genome (KT899744.1)[55]. FeatureCounts was performed using the KT89744.1 CDS or hg38 gencode.v36 as reference GFF[56]. For PolyA-selected RNA-Seq raw counts were normalized as mapped reads per million total reads per kilobase (RPKM). For Ribominus Total RNA-Seq raw counts were normalized as mapped reads per million ERCC spike-in reads per kilobase (MR/MSI/Kb).

**ChIP-Seq.** Data was first aligned using Bowtie2[57] using default settings to the human genome (hg38), and then unaligned reads were mapped to the HSV-1 strain KOS genome (KT899744.1). A modified version of the KT899744.1 genome was created, removing one copy of the repeat joint region (Δ1-9603, Δ125,845-126,977, Δ145,361-151,974). Bam files were visualized using DeepTools bamcoverage[58] with a bin size of 1 to generate bigwig files. Data was viewed in IGV viewer and exported as EPS files. For every infection performed we sequenced a paired input sample to determine the relative contributions of host and HSV-1 genome content in all conditions. Bigwig files were normalized for sequencing depth and genome quantity. This is to account for the total genomic content increase during viral replication. Host reads are multiplied by a factor to adjust for the relative down-sampling that occurs later during infection. Correspondingly viral traces are divided by a factor to adjust for the relative up-sampling, so that factor binding is relative to viral genome copy number. We have included Supplementary Table 3 as a representative dataset demonstrating the rationale behind this normalization method.

HSV-1 genome copy number was calculated based on: (i) the human genome copy number staying static during infection, (ii) known size (kbp) of the human and viral genomes and (iii) relative percentage of total reads that were mapped to the host or HSV-1 genome.

$$\left( \frac{2\ copies \times size\ of\ human\ genome \times \%\ of\ Total\ Reads\ mapped\ to\ HSV1}{\%\ of\ Total\ Reads\ mapped\ to\ host} \right) / size\ of\ HSV1\ genome$$

Scaling factors were calculated as following:

$$HSV-1\ Norm\ Factor : \left( \left( \frac{Input\ HSV-1\ mapped\ reads}{Input\ total\ reads} \right) \times Million\ IP\ Total\ Reads \right)^{-1}$$

$$Host\ Norm\ Factor : \left( \left( \frac{Input\ host\ mapped\ reads}{Input\ total\ reads} \right) \times Billion\ IP\ Total\ Reads \right)^{-1}$$

Where "Input" is the paired chromatin sample before immunoprecipitation and "IP" is the immunoprecipitated (i.e., POLR3A, POLR2A, etc.) chromatin sample.

All ChIP-Seq data is the average of 2–3 biological replicates, with paired input files. Any traces shown are the average of normalized biological replicates. Binding density heatmaps were generated using MultiBigwigSummary on normalized cellular bigwig files to all UCSC annotated mRNAs or high-confident tRNA loci from GtRNAdb[50,51]. Fold change fingerprint plots were generated using MultiBigwigSummary to create new traces for each biological replicate that are the $log_2$ fold change of infected over uninfected. Then we used deeptools computeMatrix to generate position based scores limited to regions +/− 2 kbp of transcription start sites. Deeptools plotprofile[58] calculated the average position based score across all loci evaluated. Fingerprint plots include both the average and standard deviation among biological replicate $log_2$ fold change traces.

Identification of putative A-, B-, or C-box motifs and Pol III termination signals in the HSV-1 genome was performed using FIMO[59]

**ATAC-Seq.** Data was first aligned using Bowtie2[57] and default settings to the mitochondrial genome. Unaligned reads were sequentially mapped to the human genome (hg38), and the HSV-1 strain KOS genome (KT899744.1) with the following parameters: –no-unal –local –very-sensitive-local –nodiscordant –no-mixed –contain –overlap –dovetail –phred33. Bam files were visualized using DeepTools bamcoverage[58] with a bin size of 1 to generate bigwig files. Data was viewed in IGV viewer and exported as EPS files. Cellular bigwig files were normalized for sequencing depth (excluding mitochondrial mapped reads), the y-axes values are mapped reads per billion reads. The normalized bigwig files were averaged between two biological replicates. Heatmaps and gene profiles were generated using MultiBigwigSummary and plotHeatmap[58] on normalized cellular bigwig files.

**Peak calling.** Viral peaks were called using MACS2 call peak[60], pooling treatment and control files for each condition. We first removed non-uniquely mapped sequences with SAMtools and filtered SAM or BAM for a minimum MAPQ quality score of 20[61]. Due to the small size of the viral genome (135,164 bp) we did not use the shifting model option (-nomodel), but set an extension size of 150 with a FDR of 5%. To offset the dense transcriptional landscape we used a fixed background lambda as local lambda for every peak region and a more sophisticated signal

processing approach to find subpeak summits in each enriched peak region (-call-summits). For all factors except POLR2A, subpeak summits were extended 50 bp up- and down-stream and filtered by peak strength to generate a list of high-confidence binding sites. POLR2A binding is not limited to gene promoters alone which results in broad peaks spanning entire genes, for this reason we instead used narrow peak regions called by MACS2. POLR2A peak estimates are an under-estimate due to the presence of various overlapping and nested viral transcripts. Peak sets were analyzed for intersection size and jaccard statistic using the Intervene tools Upset[62]. A summary of all peaks identified on the viral and host genome and their relative locations is provided in Supplementary Data 2 and 3.

Cellular peaks were called using MACS2[60]. We first removed non-uniquely mapped sequences with SAMtools and filtered SAM or BAM for a minimum MAPQ quality score of 20[61]. We determined the approximate extension size for each IP using MACS2 predictd, and ran MACS2 call peak for pooled samples with no shifting model (-nomodel) and an FDR of 1%. MACS2 narrow peak files were analyzed for intersection size and jaccard statistic using the Intervene tools Upset[62].

**Reporting summary.** Further information on research design is available in the Nature Research Reporting Summary linked to this article.

## Data availability

Source data are provided with this paper. The DM-tRNA-Seq data generated in this study have been deposited in the SRA database under BioProject accession: PRJNA692681 or SRA Accession: SRR13484735, SRR13484733, SRR13484734. The 4SU-Seq data generated in this study have been deposited in the SRA database under BioProject accession: PRJNA692715 or SRA Accession: SRR13484741, SRR13484740, SRR13484739, SRR13484738, SRR13484737, SRR13484736. The Ribominus RNA-Seq data generated in this study have been deposited in the SRA database under BioProject accession: PRJNA732091 or SRA Accession: SRR14632002, SRR14632001, SRR14632000, SRR14631999, SRR14631998, SRR14631997, SRR14631996, SRR14631995. The PolyA-Selected RNA-Seq data generated in this study have been deposited in the SRA database under BioProject accession: PRJNA732134 or SRA Accession: SRR14632889, SRR14632890, SRR14632888, SRR14632887, SRR14632886, SRR14632885. The ATAC-Seq data generated in this study have been deposited in the SRA database under BioProject accession: PRJNA553559 or SRA Accession: SRR10176079, SRR9661209, SRR10176081, SRR9661207. The ChIP-Seq data generated in this study have been deposited in the SRA database. BioProject accession: PRJNA553563 or SRA Accession: SRR9661306, SRR9661305, SRR9661304, SRR9661303, SRR9661302, SRR9661301. BioProject accession: PRJNA693164 or SRA Accession: SRR13484792, SRR13484791, SRR13484790, SRR13484789, SRR13484788, SRR13484787, SRR13484785, SRR13484784, SRR13484783, SRR13484786, SRR13484782, SRR13484781, SRR13484779, SRR13484777, SRR13484776, SRR13484775, SRR13484778, SRR13484774, SRR13484773, SRR13484772. BioProject accession: PRJNA732084 or SRA Accession: SRR14632140, SRR14632139, SRR14632138, SRR14632137, SRR14632143, SRR14632142, SRR14632141. BioProject accession: PRJNA732212 or SRA Accession: SRR14633311, SRR14633310, SRR14633309, SRR14633308, SRR14633307, SRR14633306, SRR14633305, SRR14633304, SRR14633303, SRR14633302, SRR14633301, SRR14633300. BioProject accession: PRJNA508787 or SRA Accession: SRR8288199, SRR8288200, SRR8288191, SRR8288192.

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

## Acknowledgements

Thanks to Hannah Fox for datasets and Jill Dembowski for thoughtful discussions. NIH grants R01-AI030612 and R21-AI156065 to N.A.D. NIH grants T32-AI060525 and F31-AI36251 to S.E.D. American Cancer Society Postdoctoral Award 131370-PF-17-245-01-MPC to J.M.T. NIH grant R01-AI147183 to B.A.G. B.A.G. is an HHMI Investigator.

## Author contributions

The conceptualization of this study was done by S.E.D. and N.A.D. Data curation was performed by S.E.D. Funds to conduct the study were provided by S.E.D. and N.A.D. The experiments in this study were performed by S.E.D., F.L.S. and N.A.D. Methodology was provided by S.E.D., F.L.S., J.M.T. and N.A.D. The original draft of the manuscript was composed by S.E.D. Subsequent writing, review and editing of the manuscript was performed by S.E.D., J.M.T., B.A.G. and N.A.D.

## Competing interests

The authors declare no competing interests.
