## [Peer Review File · Nature Communications]

Reviewer #1 (Remarks to the Author):

In this article, the authors analyze how HSV-1 perturbs Pol III transcription using DM-RNA-Seq, 4SU-Seq, ChIP-Seq, and ATAC-Seq. While this provides potential interesting and important insights into HSV-1 infection, there are a number of problems with the analysis and the presentation in the manuscript that need to be addressed. In particular, some of the analyses appear to be rather ad hoc and not properly quantitative to allow the conclusions drawn from them. Furthermore, important details of the bioinformatics analysis remain unclear and the authors do not always use state-of-art methods for NGS data analysis.

Detailed comments:

1) It is not specified which program was used for mapping 4SU-Seq and DM-tRNA-Seq data (see line 829ff.). The authors state that they used the public server at usegalaxy.org to analyze the data, however this provides multiple methods for sequencing read mapping, thus this needs to be specified. Below they state that they used HISAT2 for mapping Ribominus Total or PolyA-selected RNA-Seq data, but it is not clear whether the same or a different approach was used for 4SU-Seq and DM-tRNA-Seq data. If a different approach was used, why? The same approach (mapping against host, viral genome and rRNA sequences) could (and should) be used for all types of RNA-seq data. Furthermore, it seems that only a part of the tRAX pipeline was used ("Mature and premature-tRNA assemblies for hg38 were generated"), but from the description of the tRAX pipeline, which includes mapping, abundance estimation and differential gene expression analysis, it remains unclear which part was used here. Does this only include the compilation of the tRNA Reference DB? This needs to be clarified. Also it is not quite clear what is done with these assemblies as mapping is only performed against the pre-tRNA sequences. Is this only done for identifying which tRNA regions need to be masked in hg38?

2) It is not specified which method was used for differential gene expression analysis, i.e. calculation of fold-changes and p-values. Again usegalaxy.org provides multiple alternative state-of-the-art methods (limma, DESeq2, edgeR). The figure legends suggest that they simply calculated fold-changes from normalized expression values and FDR using some sort of statistical test and some multiple testing correction (Benjamin and Hochberg, maybe?). This needs to be specified. Furthermore, if this approach was used, this is *not* the state-of-the-art and should be performed with proper methods for count-based differential gene expression analysis such as limma, DESeq2, edgeR. Notably, at least DESeq2 and edgeR can use spike-in read numbers for normalization.

3) The authors state that "FeatureCounts was performed using the KT89974.1 CDS as the reference GFF" to count reads for viral genes, but it is not specified how reads for host protein-coding genes (in particular Pol II and III subunits, see Supplemental Fig. 4-2) are counted, in particular based on which annotation. For calculating normalized expression values in PolyA-selected RNA-Seq, they also use a "novel" metric, i.e. mapped reads per billion total reads per kilobase (MR/BTR/Kb). Commonly used metrics include mapped reads per million total reads per kilobase (RPKM). If I understand their measure correctly, it is essentially RPKM but with mapped reads being divided by billion instead of million. This would simply multiply all values by 1000. I am not quite sure if this is actually what they did as the values in Supplemental Fig. 4-2 B,C (the only place where this is actually used) appear to be quite small, i.e. in the range of 0 to 10/10⁴. Or should this be 0-100,000? In the latter case, using RPKM would be more reasonable as it would give values in the range of 0 to 100, which are reasonable RPKM values for human genes and values readers will be familiar with. Ribominus Total RNA-Seq raw counts were normalized as mapped reads per million ERCC spike-in reads per kilobase, which is more like the standard RPKM measure, however they use two alternative names for this measure. In the methods it is denoted as MR/MSI/Kb, while in the Figures, it is denoted as Norm MR. Only one name should be used throughout the paper.

4) Regarding the discrimination of pre-tRNAs and mature-tRNAs: The authors claim that with DM-tRNA-Seq, they "we can accurately discriminate between pre and mature-tRNA species and

between the approximately 500 different tRNA-encoding loci within the host genome". While reads mapping only to pre-tRNA sequences can be uniquely identified as originating from pre-tRNAs, reads mapping to regions of mature tRNAs could also be originating from pre-tRNAs. From the description in the methods it appears that reads were mapped first against the mature tRNA sequences and thus all these reads are assigned to mature tRNAs and only unaligned reads would then be aligned against pre-tRNA sequences. Thus, essentially only reads mapping to introns and the 5' and 3' leader sequences would be counted for pre-tRNAs while all "exonic" reads would be counted towards the mature tRNAs. While I do not see an alternative method to distinguish reads for mature and pre-tRNAs, this issue needs to be highlighted in the results section, i.e. that expression values and fold-changes for mature tRNAs also include reads originating from pre-tRNAs.

5) Similarity of transcription factor binding to POLR3A is determined using PCA on "GTF binding profiles for the viral genome" (by the way the abbreviation GTF is not really defined. It does not only appear to include the different GTF genes) which were determined by binning the normalized mapped read traces in 10bp bins. Determining the distance of points based only the first two principle components of the PCA is a very crude and imprecise way of measuring the similarity of transcription factors binding. At the very least, the similarity/distance based on all bins should be evaluated, e.g. using correlation coefficients or Euclidean distances. Furthermore, even in the PCA plots POLR3A is not even particular close or closest to POLR2A, thus I find the conclusion that "POLR3A binding most closely resembled POLR2A" rather bold if not actually incorrect. With the exception of KOS 2h, POLR3A is actually closer to TBP in the PCA plot, and at individual time points other transcription factors are even closer.

6) In Fig. 6 they defined both Pol II coincident and Pol II independent binding but this appears to have been done manually in an ad hoc way and not in any well-defined/quantitative manner. They furthermore claim "These binding contexts are drastically different from those characterized (19, 20) and observed in our own data for host promoters", however the analysis this is based on is quite different from the one they performed on the viral genome. For the host promoters, they at least performed a semi-quantitative analysis where one can see the binding of the different transcription factors relative to the different types of promoters. At the very least a similar analysis should be performed for the viral genome, e.g. by first identifying POLR3A peaks using peak calling with GEM, MACS2 or similar and then visualizing the binding profile relative to those peaks. Otherwise, this conclusion cannot stand. Furthermore, they should develop some objective/quantitative criteria for classifying binding sites of Pol III as either Pol II coincident and Pol II independent.

7) Other comments:

- line 253f: "We observed increased recruitment of POLR2A to tRNA loci for all viruses tested, with the exception d109": A small increase within tRNA loci can also be seen for d109. Mean and standard deviation are above the zero line. This needs to be rephrased.
- line 257: "The absence of Pol II recruitment to tRNA loci after d109 infection agrees with our earlier findings that d109 infection does not cause tRNA upregulation". As noted above, there is some recruitment albeit much lower than for other factors. Nevertheless, this sentence needs to be adjusted.
- line 271f. "These cellular enhancers": The use of the term "enhancer" is confusing here as enhancers are commonly defined as "a short (50–1500 bp) region of DNA that can be bound by proteins (activators)" [Wikipedia] and this is not what the authors mean here. This needs to be rewritten.
- line 276: "Ultimately these results do not explain enhanced Pol II recruitment to tRNA in an environment where their transcription and protein expression levels are globally decreased." It is not completely clear what their refers to. Please rephrase.
- It is not clear to me why the authors also performed PolyA-selected RNA-Seq. This is only used for quantifying expression of subunits for Pol II and III (in Supplemental Fig. 4-2). This could also be done with total RNA-seq for which they also used ERCC spike-ins which allows absolute quantification of gene expression.

8) Figures/Figure legends:

- Fig. 2A: It is unclear what the bars are showing. Mean/median over different tRNAs? What is indicated by error bars?

- 2C: Since the data points for pre-tRNAs are overlaying the points for mature tRNAs, one cannot really appreciate the effect for mature tRNAs, thus separate figures should be shown for these. It would also be better to show scatter plots of log₂ FC between n12 and KOS. This would allow evaluating how similar changes are between these two.
- 2E: The color scale makes it difficult to decipher whether genes are slightly up- or down-regulated as grey and blackish/green are difficult to distinguish. Please use a red-blue scale.
- 2E: Why are fold-changes only shown for DM-tRNA but Norm MR only for the 4SU data. Furthermore, since no spike-ins were used for the 4SU data, how did they calculate Norm MR values here?
- Fig. 3D: "Differentially expressed transcripts are highlighted with asterisks." Up- or down-regulated? Which fold-change was required as cutoff? Which p-value? Do these genes have to be differentially expressed for all comparisons against hp66? Please clarify.
- Fig. 4A: "The average log₂ fold change for infected over the matched uninfected": Were fold-changes calculated per position or per bins?
- Fig. 4A: Standard deviation as in Supplementary Figure 4-1 should be shown for all transcription factors, at least as a Supplementary Figure.
- Fig. 5: "viral genome copy number": How was that determined?

In general, Figure legends and bioinformatics/statistical methods would benefit from some more details.

Reviewer #2 (Remarks to the Author):

In their manuscript entitled "Manipulation of RNA polymerase III by Herpes Simplex Virus-1" the authors report two major interesting findings. 1) HSV-1 stimulates tRNA transcription by 10-fold resulting in a 2-fold increase in total tRNA levels within 12 hpi. Of note, tRNA induction does not appear to rely on a single viral gene but rather the extent of viral transcription itself. 2) Pol III and associated factors bind the HSV genome suggesting a previously unrecognized role of Pol III in HSV-1 gene expression. The work is based on an extensive set of big data which appear well controlled.

Major comments:

1. The first key finding is well documented by a large set of data. Based on the analysis of an array of viral mutants and conditions, the authors conclude that not the expression of a single viral gene but rather the extent of viral gene transcription itself and possible cellular responses to their aberrant nature triggers tRNA upregulation. While this conclusion seems reasonable based on the presented data it would be important to include all Northern Blots that were performed into the Supplements and properly quantify the blots including a statistical analysis to provide statistically solid data. It looks to me that d92 (Δ ICP4/ICP27) shows hardly any tRNA induction.
2. The second key finding is based on an extensive set of ChIP data. These provide good evidence of Pol III recruitment to the viral genome. However, functional recruitment of Pol III would require direct confirmation of Pol III-derived transcripts, e.g. by sequencing of Pol III-derived RNA similar to NET-seq. This is probably beyond the scope of this manuscript but will sure be very interesting in the future. Saying this, I do find it hard to believe that this represents active Pol III transcription as POLR3A ChIP data at least in parts fully overlaps with known viral genes (as observable for ICP27 in Fig. 5A). The authors should look for known promoter elements and potentially terminator elements in the viral genome and correlate these to their best candidate regions for Pol III transcription. Pol III promoter elements are quite well defined. It would be of particular interest whether some of the HSV-1 miRNAs might originate from Pol III transcripts.
3. While the paper itself is well written, the figure legends are kept rather brief and it took me a

while to understand what is shown on the figures, e.g. Fig.2A,B, Fig 4B, and how they were generated. Additional information on what is shown in all (!) figures would be very helpful. In many cases, abbreviations are not explained. Even for someone familiar with the employed HSV-1 mutants, it is difficult to follow which viral genes they lack. Each figure or figure legend should thus somewhere contain the information which genes the respective mutants are lacking (either in the figure itself or in the legend). A good example of this is Suppl. Fig. 3-1 which includes a great variety of different mutant but no information on them. One thus needs to go and find what each mutant is elsewhere. The same is true for the main text. It would be very helpful if a uniform way of presentation would be used, i.e. Δ ICP4 (n12) rather than simply n12. Along the same lines, the hp66 mutant is first introduced in the main results section without any information on the virus. The readers need to go to the respective old papers to understand what was used.

Minor comments:

Fig. 3. How many biological replicates were performed? It simply states here: "Data is the average of biological replicates..."

Fig. 3B "Representative Northern Blots" => How many replicates were done? Here, Hp66 lacks a band for the Isoleucine (Ile) tRNA but otherwise shows similar levels as others. Additional replicates performed should be shown and quantified.

Please include some basic information into the results section on the principle of DM-tRNA-seq.

What is this and how is it done? Even the Methods part only refers to other papers.

Line 240-241: Hard to understand, please rephrase.

Line 327: Grammar: "This factor has very select host binding..."

Reviewer #3 (Remarks to the Author):

The manuscript by Dremel et al., reports the very interesting finding that infection with herpes simplex virus 1 (HSV-1) perturbs the RNA Pol III landscape in that tRNA expression is increased ~10 fold after infection. The authors perform a comprehensive characterization of the Pol III landscape including DM-RNA-Seq, 4SU-Seq, ChIP-Seq, and ATAC-Seq. They also use a number of well characterized HSV-1 viral mutants to determine how viral infection causes the induction of Pol III tRNAs. This does not however lead to specific viral gene products or stages in the viral lifecycle as clear causes of this induction. Instead this leads to elimination of some of the promising suspects, that is, synthesis of specific viral transcripts, nascent viral genomes or viral progeny are not required. Further host tRNAs with a specific codon bias were not targeted but increased transcription was seen in euchromatin, actively transcribed loci, leading the authors to conclude that tRNA upregulation is linked to unique crosstalk between Pol II and Pol III transcriptional machinery. The authors also report that Pol III binds the HSV-1 genome, which leads the authors to speculate (lines 418 and 419) that they provide the first evidence for a putative Pol III transcript from the HSV-1 genome. However, such a transcript is not shown in this report and this remains speculation.

Overall, the experiments are performed very well and the data are comprehensive. The findings are novel for HSV-1, which has not previously been shown to induce Pol III tRNAs. Although the manuscript is generally well written, I do have several concerns that the authors should address to strengthen the manuscript and to make it more comprehensible to a wider range of readers.

1. Although the viruses used in the study are somewhat described in the Materials and Methods, a more complete description starting on line 113 would be beneficial to the non HSV-1 virologist reader. For example, it could be stated that d109 has all five IE genes deleted and although it enters the nucleus, there is no viral gene expression. Deletion of ICP4 restricts viral gene expression to IE genes. Line 195, hp66 harbors a mutation in the viral DNA polymerase gene.
2. In a number of instances, especially in the Discussion, the authors pose a hypothesis to explain their findings but there is no direct evidence presented here and thus they state that "further work needs to be done". A discussion should include speculation and point to future studies but in the end there is a lot of speculation and it is difficult to come up with firm conclusions.
3. There is a confusing section in the Discussion that appears to mix up mRNA and protein. Starting on line 395, the authors describe that productive viral infection takes a toll on host cell homeostasis including the unfolded protein response UPR. Line 398 it is stated that "HSV-1 transcripts are composed from a high density of intrinsically disordered domains and due to rapid rate of synthesis are prone to misfolding and aggregation traits that robustly trigger a UPR. Proteins not mRNA transcripts can have intrinsically disordered domains and can under some

conditions lead to misfolding and aggregation and this can lead to the Unfolded Protein Response.

4. Lines 418 ad 419, there was no evidence presented in this study for a putative Pol III transcript derived from the HSV-1.

5. I would suggest the discussion be tightened as it somewhat rambles among the different findings that were observed and the possible hypotheses to explain the findings, to discounting several of these or stating that more work is needed.

REVIEWER COMMENTS

Reviewer #1 (Remarks to the Author):

In this article, the authors analyze how HSV-1 perturbs Pol III transcription using DM-RNA-Seq, 4SU-Seq, ChIP-Seq, and ATAC-Seq. While this provides potential interesting and important insights into HSV-1 infection, there are a number of problems with the analysis and the presentation in the manuscript that need to be addressed. In particular, some of the analyses appear to be rather ad hoc and not properly quantitative to allow the conclusions drawn from them. Furthermore, important details of the bioinformatics analysis remain unclear and the authors do not always use state-of-art methods for NGS data analysis.

Detailed comments:

1) It is not specified which program was used for mapping 4SU-Seq and DM-tRNA-Seq data (see line 829). The authors state that they used the public server at usegalaxy.org to analyze the data, however this provides multiple methods for sequencing read mapping, thus this needs to be specified. Below they state that they used HISAT2 for mapping Ribominus Total or PolyA-selected RNA-Seq data, but it is not clear whether the same or a different approach was used for 4SU-Seq and DM-tRNA-Seq data. If a different approach was used, why? The same approach (mapping against host, viral genome and rRNA sequences) could (and should) be used for all types of RNA-seq data. Furthermore, it seems that only a part of the tRAX pipeline was used ("Mature and premature-tRNA assemblies for hg38 were generated"), but from the description of the tRAX pipeline, which includes mapping, abundance estimation and differential gene expression analysis, it remains unclear which part was used here. Does this only include the compilation of the tRNA Reference DB? This needs to be clarified. Also it is not quite clear what is done with these assemblies as mapping is only performed against the pre-tRNA sequences. Is this only done for identifying which tRNA regions need to be masked in hg38?

Response: Our mapping strategy for DM-tRNA-Seq and 4SU-seq has been more clearly described in the Bioinformatic Analysis subsection of the methods, see lines 601-644.

2) It is not specified which method was used for differential gene expression analysis, i.e. calculation of fold-changes and p-values. Again usegalaxy.org provides multiple alternative state-of-the-art methods (limma, DESeq2, edgeR). The figure legends suggest that they simply calculated fold-changes from normalized expression values and FDR using some sort of statistical test and some multiple testing correction (Benjamin and Hochberg, maybe?). This needs to be specified. Furthermore, if this approach was used, this is *not* the state-of-the-art and should be performed with proper methods for count-based differential gene expression analysis such as limma, DESeq2, edgeR. Notably, at least DESeq2 and edgeR can use spike-in read numbers for normalization.

Response: We did use edgeR with multiple testing correction (Benjamini and Hochberg) for DE analysis, and have included specific details on this analysis in the methods section at lines 625-627.

3) The authors state that "FeatureCounts was performed using the KT89974.1 CDS as the reference GFF" to count reads for viral genes, but it is not specified how reads for host protein-coding genes (in particular Pol II and III subunits, see Supplemental Fig. 4-2) are counted, in particular based on which annotation.

Response: We used the most recent version of gencode (.v36) as a GFF in our quantification. We have expanded on our mapping strategy for RNA-Seq within the Bioinformatic Analysis subsection of the methods, see lines 649-650.

For calculating normalized expression values in PolyA-selected RNA-Seq, they also use a "novel" metric, i.e. mapped reads per billion total reads per kilobase (MR/BTR/Kb). Commonly used metrics

include mapped reads per million total reads per kilobase (RPKM). If I understand their measure correctly, it is essentially RPKM but with mapped reads being divided by billion instead of million. This would simply multiply all values by 1000. I am not quite sure if this is actually what they did as the values in Supplemental Fig. 4-2 B,C (the only place where this is actually used) appear to be quite small, i.e. in the range of 0 to 10/10⁴. Or should this be 0-100,000? In the latter case, using RPKM would be more reasonable as it would give values in the range of 0 to 100, which are reasonable RPKM values for human genes and values readers will be familiar with.

Response: Our prior normalization was identical to RPKM, except multiplied by 1000. However, for ease of understanding the data we have shifted values to RPKM.

Ribominus Total RNA-Seq raw counts were normalized as mapped reads per million ERCC spike-in reads per kilobase, which is more like the standard RPKM measure, however they use two alternative names for this measure. In the methods it is denoted as MR/MSI/Kb, while in the Figures, it is denoted as Norm MR. Only one name should be used throughout the paper.

Response: Figure legends and labels have been updated to only use MR/MSI/KB.

4) Regarding the discrimination of pre-tRNAs and mature-tRNAs: The authors claim that with DM-tRNA-Seq, they "we can accurately discriminate between pre and mature-tRNA species and between the approximately 500 different tRNA-encoding loci within the host genome". While reads mapping only to pre-tRNA sequences can be uniquely identified as originating from pre-tRNAs, reads mapping to regions of mature tRNAs could also be originating from pre-tRNAs. From the description in the methods it appears that reads were mapped first against the mature tRNA sequences and thus all these reads are assigned to mature tRNAs and only unaligned reads would then be aligned against pre-tRNA sequences. Thus, essentially only reads mapping to introns and the 5' and 3' leader sequences would be counted for pre-tRNAs while all "exonic" reads would be counted towards the mature tRNAs. While I do not see an alternative method to distinguish reads for mature and pre-tRNAs, this issue needs to be highlighted in the results section, i.e. that expression values and fold-changes for mature tRNAs also include reads originating from pre-tRNAs.

Response: We present three lines of evidence that support accurate quantification of mature vs. premature-tRNA reads:

1. The DM-tRNA-Seq method is used because it facilitates full processive reverse transcription, allowing mapping of full length tRNA species—thus, we expect very few reads assigned to mature-tRNA species to have originated from premature tRNA truncated products. We have also added additional lines to the methods section describing size based analysis of reads mapped to mature- and premature-tRNA assemblies and how this gave increased confidence in mapping of mature vs. premature tRNA species.
2. Additionally mature-tRNA levels are ~1000-fold higher than premature-tRNA levels, so any potential misassignment of premature reads would not significantly impact quantification of mature levels. Rather the inverse would be true, pre-tRNA shifts may be an underestimate. We have updated our methods section to convey this caveat, "As our pipeline maps sequentially to mature- and then premature-tRNA, the amount of pre-tRNA reads quantified will be an underestimate. Additionally due to high sequence overlap between tRNA genes for the same isodecoder, unless the premature reads contain locus specific flanking regions or introns these reads will be split between genes encoding the same isodecoder."
3. Finally, for this reason we performed northern blot analysis of mature- and premature-tRNA species to validate our sequencing approach. Since northern blot analysis allows for determination of species size and is a direct RNA measurement this orthogonal approach increased confidence in our sequencing results.

5) Similarity of transcription factor binding to POLR3A is determined using PCA on "GTF binding profiles for the viral genome" (by the way the abbreviation GTF is not really defined. It does not only appear to include the different GTF genes) which were determined by binning the normalized mapped

read traces in 10bp bins. Determining the distance of points based only the first two principle components of the PCA is a very crude and imprecise way of measuring the similarity of transcription factors binding. At the very least, the similarity/distance based on all bins should be evaluated, e.g. using correlation coefficients or Euclidean distances. Furthermore, even in the PCA plots POLR3A is not even particular close or closest to POLR2A, thus I find the conclusion that "POLR3A binding most closely resembled POLR2A" rather bold if not actually incorrect. With the exception of KOS 2h, POLR3A is actually closer to TBP in the PCA plot, and at individual time points other transcription factors are even closer.

Response: Per reviewer's suggestion we have updated our figure to Pearson correlation analysis of 10bp bins spanning the entire viral genome for all factors. We still find that early during infection, POLR3A binding mostly closely resembled that of POLR2A. A similar analysis was performed for host data—with the exception that correlation analysis was limited to either Pol III-dependent genes, or Pol II-dependent promoters. We have also performed peak analysis for factor binding on the viral and host genome and used jaccard and intersection statistics to characterize binding phenotypes. Using these two new metrics we have stastically quantified Pol III transcription factor binding in all samples, and compared phenotypes between the host and virus. Our prior conclusion holds true, namely that POLR3A binding to the viral genome strongly mimicks that of POLR2A.

6) In Fig. 6 they defined both Pol II coincident and Pol II independent binding but this appears to have been done manually in an ad hoc way and not in any well-defined/quantitative manner. They furthermore claim "These binding contexts are drastically different from those characterized (19, 20) and observed in our own data for host promoters", however the analysis this is based on is quite different from the one they performed on the viral genome. For the host promoters, they at least performed a semi-quantitative analysis where one can see the binding of the different transcription factors relative to the different types of promoters. At the very least a similar analysis should be performed for the viral genome, e.g. by first identifying POLR3A peaks using peak calling with GEM, MACS2 or similar and then visualizing the binding profile relative to those peaks. Otherwise, this conclusion cannot stand. Furthermore, they should develop some objective/quantitative criteria for classifying binding sites of Pol III as either Pol II coincident and Pol II independent.

Response: As discussed above in our reponse to comment #4, we have now included MACs based peak-calling to compare the viral and host genome. The degree of overlap was quantified using peak calling algorithms and Pearson correlation scores using position based bin analysis. We have still included traces visualizing factor binding to the entire viral genome, this is necessary because ChIP-Seq bioinformatics tools were never intended to be used on such a transcriptionally dense assembly. The HSV-1 and host genomes have an approximate coding capacity of 0.6 and ~0.01 genes per kbp, respectively. For this reason, we have included statistical analysis as well as traces as they both provide essential information.

7) Other comments:

- line 253f: "We observed increased recruitment of POLR2A to tRNA loci for all viruses tested, with the exception d109": A small increase within tRNA loci can also be seen for d109. Mean and standard deviation are above the zero line. This needs to be rephrased.

Response: We have updated the text to the following: "We observed increased recruitment of POLR2A to tRNA loci during Δ ICP4, Δ ICP27, Δ ICP22, and wildtype HSV-1 infection. Of the mutants tested, Δ ICP0/4/22/27/47 infection had the lowest level of POLR2A recruitment to tRNA loci (Supplementary Fig. 8)."

- line 257: "The absence of Pol II recruitment to tRNA loci after d109 infection agrees with our earlier findings that d109 infection does not cause tRNA upregulation". As noted above, there is some recruitment albeit much lower than for other factors. Nevertheless, this sentence needs to be adjusted.

Response: We have updated the text to the following: "Minimal Pol II recruitment to tRNA loci after Δ ICP0/4/22/27/47 infection agrees with our earlier findings that this mutant does not cause tRNA upregulation (Fig. 1 and 3)."

- line 271f. "These cellular enhancers": The use of the term "enhancer" is confusing here as enhancers are commonly defined as "a short (50–1500 bp) region of DNA that can be bound by proteins (activators)" [Wikipedia] and this is not what the authors mean here. This needs to be rewritten.

Response: We have updated the text to use the term "activator" instead of "enhancer".

- line 276: "Ultimately these results do not explain enhanced Pol II recruitment to tRNA in an environment where their transcription and protein expression levels are globally decreased." It is not completely clear what their refers to. Please rephrase.

Response: We have rephrased the text to the following: "These results rule out a model in which increased Pol III machinery abundance promotes increased tRNA transcriptional output."

- It is not clear to me why the authors also performed PolyA-selected RNA-Seq. This is only used for quantifying expression of subunits for Pol II and III (in Supplemental Fig. 4-2). This could also be done with total RNA-seq for which they also used ERCC spike-ins which allows absolute quantification of gene expression.

Response: We have also included Total RNA-Seq data for Pol II/III factors to complement our PolyA Selected RNA-Seq analysis. Both datasets provide valuable information regarding the shifting transcriptional landscape during infection. However, our prior conclusion is true in both sequencing methods, namely that HSV-1 downregulates transcription of the Pol II/III machinery.

8) Figures/Figure legends:

- Fig. 2A: It is unclear what the bars are showing. Mean/median over different tRNAs? What is indicated by error bars?

Response: We have added this information to the figure legend. Data bars are average, error bars are standard deviation and individual data points are experimental replicates.

- 2C: Since the data points for pre-tRNAs are overlaying the points for mature tRNAs, one cannot really appreciate the effect for mature tRNAs, thus separate figures should be shown for these. It would also be better to show scatter plots of log₂ FC between n12 and KOS. This would allow evaluating how similar changes are between these two.

Response: We have included these plots as a new supplemental figure (Supplementary Fig. 2).

- 2E: The color scale makes it difficult to decipher whether genes are slightly up- or down-regulated as grey and blackish/green are difficult to distinguish. Please use a red-blue scale.

Response: We have changed the color scale to make it easier for readers.

- 2E: Why are fold-changes only shown for DM-tRNA but Norm MR only for the 4SU data. Furthermore, since no spike-ins were used for the 4SU data, how did they calculate Norm MR values here?

Response: Compared to DM-tRNA-Seq, 4SU-Seq has a limited level of sensitivity in measuring tRNA species. For this reason we thought it more transparent to plot normalized mapped reads, rather than fold change.

In regards to normalization, we did not deplete our 4SU pulldown samples of rRNA. As rRNA levels do not shift during HSV-1 infection, it served as a surrogate to a spike-in control (MR/Billion rRNA MR/KB). Thus our rRNA data was normalized as mapped reads per billion rRNA reads per kbp. We already had this information within our methods section but have made it more prominent, see lines 641-644.

- Fig. 3D: "Differentially expressed transcripts are highlighted with asterisks." Up- or down-regulated? Which fold-change was required as cutoff? Which p-value? Do these genes have to be differentially expressed for all comparisons against hp66? Please clarify.

Response: We have updated the figure legend to clarify: "Transcripts highlighted with asterisks are upregulated (\log_2 fold change > 0) in all mutants relative to Δ UL30 infection."

- Fig. 4A: "The average \log_2 fold change for infected over the matched uninfected": Were fold-changes calculated per position or per bins?

Response: Fold change calculations were bin based. Our quantification strategy for ChIP-Seq has been more clearly described in the Bioinformatic Analysis subsection of the methods, 676-682

- Fig. 4A: Standard deviation as in Supplementary Figure 4-1 should be shown for all transcription factors, at least as a Supplementary Figure.

Response: We have added a new supplemental figure (Supplementary Fig. 7) which plots the mean and standard deviation for ChIP-Seq signal at mRNA and tRNA promoters.

- Fig. 5: "viral genome copy number": How was that determined?

Response: We have expanded on this calculation in the ChIP-Seq Bioinformatic Analysis subsection of the methods, see lines 661-671.

In general, Figure legends and bioinformatics/statistical methods would benefit from some more details.

Reviewer #2 (Remarks to the Author):

In their manuscript entitled “Manipulation of RNA polymerase III by Herpes Simplex Virus-1” the authors report two major interesting findings. 1) HSV-1 stimulates tRNA transcription by 10-fold resulting in a 2-fold increase in total tRNA levels within 12 hpi. Of note, tRNA induction does not appear to rely on a single viral gene but rather the extent of viral transcription itself. 2) Pol III and associated factors bind the HSV genome suggesting a previously unrecognized role of Pol III in HSV-1 gene expression. The work is based on an extensive set of big data which appear well controlled.

Major comments:

1. The first key finding is well documented by a large set of data. Based on the analysis of an array of viral mutants and conditions, the authors conclude that not the expression of a single viral gene but rather the extent of viral gene transcription itself and possible cellular responses to their aberrant nature triggers tRNA upregulation. While this conclusion seems reasonable based on the presented data it would be important to include all Northern Blots that were performed into the Supplements and properly quantify the blots including a statistical analysis to provide statistically solid data. It looks to me that d92 (Δ ICP4/ICP27) shows hardly any tRNA induction.

Response: We have updated the supplemental materials or main body of the paper to include quantification of all northern blots referenced. Individual biological replicates are visible as symbols, columns are the data average, and error bars are standard deviation. Of mutants tested d92 has an intermediate phenotype, there is some upregulation of tRNA compared to mock, but certainly not as extensive as wildtype HSV-1 infection induces. Since d92 is deleted for two immediate early genes with critical functions in promoting viral transcription we would expect this virus to synthesize very few viral transcripts, fitting with our hypothesis regarding amount, not identity, stimulating tRNA synthesis.

2. The second key finding is based on an extensive set of ChIP data. These provide good evidence of Pol III recruitment to the viral genome. However, functional recruitment of Pol III would require direct confirmation of Pol III-derived transcripts, e.g. by sequencing of Pol III-derived RNA similar to NET-seq. This is probably beyond the scope of this manuscript but will sure be very interesting in the future. Saying this, I do find it hard to believe that this represents active Pol III transcription as POLR3A ChIP data at least in parts fully overlaps with known viral genes (as observable for ICP27 in Fig. 5A). The authors should look for known promoter elements and potentially terminator elements in the viral genome and correlate these to their best candidate regions for Pol III transcription. Pol III promoter elements are quite well defined. It would be of particular interest whether some of the HSV-1 miRNAs might originate from Pol III transcripts.

Response: We agree that that large bulk of Pol III binding is coincident with Pol II and unlikely to result in a Pol III-dependent transcript—this result has been expanded on further in Fig. 5. For Pol II-independent binding events, we have updated Figure 6 to show the position of sequences matching, A, B, or C-box motifs as well as Pol III termination signals (TTTT).

3. While the paper itself is well written, the figure legends are kept rather brief and it took me a while to understand what is shown on the figures, e.g. Fig.2A,B, Fig 4B, and how they were generated. Additional information on what is shown in all (!) figures would be very helpful. In many cases, abbreviations are not explained. Even for someone familiar with the employed HSV-1 mutants, it is difficult to follow which viral genes they lack. Each figure or figure legend should thus somewhere contain the information which genes the respective mutants are lacking (either in the figure itself or in the legend). A good example of this is Suppl. Fig. 3-1 which includes a great variety of different mutant but no information on them. One thus needs to go and find what each mutant is elsewhere. The same is true for the main text. It would be very helpful if a uniform way of presentation would be used, i.e. Δ ICP4 (n12) rather than simply n12. Along the same lines, the hp66 mutant is first introduced in the main results section without any information on the virus. The readers need to go to

the respective old papers to understand what was used.

Response: All references to mutant HSV-1 viruses in the text and figures have been changed to the format of Δ protein or WT, to aid readers.

See additional text to clarify mutants at:

“During infection with d109, the virus enters the nucleus but fails to synthesize any nascent viral proteins or viral genomes; however, it robustly stimulates a cGAS-mediated innate immune response²³. Δ ICP4 infection overproduces IE transcripts but is deficient in the synthesis of early (E) and late (L) viral proteins, nascent viral genomes, and viral progeny²⁴.”

“Viruses used include: Δ ICP4 (n12), Δ ICP27 (5dl1.2), Δ ICP0 (n212), Δ ICP22 (n199), Δ ICP0/4/22/27/47 (d109), Δ UL30 (hp66) or wildtype (KOS) HSV-1. ICP4, ICP27, ICP0, and ICP22 are viral IE proteins with various roles in promoting viral gene expression. UL30 is the viral DNA polymerase required for genome replication.”

We have also added a table in the methods section clearly listing all mutants used in the study.

Virus	Mutant Genotype	Background Strain
n199 ⁴¹	ICP22 nonsense	KOS
n212 ⁴²	ICP0 nonsense	KOS
d99 ⁴³	ICP0 deletion	KOS
n12 ²⁴	ICP4 nonsense	KOS
d120 ²⁴	ICP4 deletion	KOS
5dl1.2 ⁴⁴	ICP27 deletion	KOS
d92 ⁴⁵	ICP4/27 deletion	KOS
d109 ⁴³	ICP0/4/22/27/47 deletion	KOS
R3616 ⁴⁶	RL1 deletion	F
F- Δ ICP47 ⁴⁷	ICP47 deletion	F
F- Δ ICP47-R ⁴⁷	ICP47 deletion revertant	F
hp66 ⁴⁸	UL30 deletion	KOS
KOS ⁴⁹	Wildtype	N/A
F-Strain	Wildtype	N/A

Minor comments:

Fig. 3. How many biological replicates were performed? It simply states here: “Data is the average of biological replicates....”

Response: At least two biological replicates were performed for all conditions. We have included new supplemental figures with quantifications for all northern blots, these include individual data points representative of biological replicates.

Fig. 3B “Representative Northern Blots” => How many replicates were done? Here, Hp66 lacks a band for the Isoleucine (Ile) tRNA but otherwise shows similar levels as others. Additional replicates performed should be shown and quantified.

Response: Similar to comment above, we have included new supplemental figures with quantifications for all northern blots, these include individual data points representative of biological replicates.

Please include some basic information into the results section on the principle of DM-tRNA-seq. What is this and how is it done? Even the Methods part only refers to other papers.

Response: We have included additional text within the results and methods section to explain the principles and advantages behind DM-tRNA-Seq.

Line 240-241: Hard to understand, please rephrase.

Response: We have rephrased this sentence to the following: “tRNA loci with increased POLR2A recruitment were found in accessible regions of the genome (Fig. 4B). These same tRNA loci were upregulated after infection in our DM-tRNA-Seq dataset (Fig. 4B-C).”

Line 327: Grammar: “This factor has very select host binding...”

Response: We have rephrased this sentence to the following: “BRF2 binds a very small repertoire of host promoters (around a dozen targets¹¹), an observation consistent with our own analysis of host binding (Supplementary Fig. 14, Fig. 6B).”

Reviewer #3 (Remarks to the Author):

The manuscript by Dremel et al., reports the very interesting finding that infection with herpes simplex virus 1 (HSV-1) perturbs the RNA Pol III landscape in that tRNA expression is increased ~10 fold after infection. The authors perform a comprehensive characterization of the Pol III landscape including DM-RNA-Seq, 4SU-Seq, ChIP-Seq, and ATAC-Seq. They also use a number of well characterized HSV-1 viral mutants to determine how viral infection causes the induction of Pol III tRNAs. This does not however lead to specific viral gene products or stages in the viral lifecycle as clear causes of this induction. Instead this leads to elimination of some of the promising suspects, that is, synthesis of specific viral transcripts, nascent viral genomes or viral progeny are not required. Further host tRNAs with a specific codon bias were not targeted but increased transcription was seen in euchromatin, actively transcribed loci, leading the authors to conclude that tRNA upregulation is linked to unique crosstalk between Pol II and Pol III transcriptional machinery. The authors also report that Pol III binds the HSV-1 genome, which leads the authors to speculate (lines 418 and 419) that they provide the first evidence for a putative Pol III transcript from the HSV-1 genome. However, such a transcript is not shown in this report and this remains speculation.

Overall, the experiments are performed very well and the data are comprehensive. The findings are novel for HSV-1, which has not previously been shown to induce Pol III tRNAs. Although the manuscript is generally well written, I do have several concerns that the authors should address to strengthen the manuscript and to make it more comprehensible to a wider range of readers.

1. Although the viruses used in the study are somewhat described in the Materials and Methods, a more complete description starting on line 113 would be beneficial to the non HSV-1 virologist reader. For example, it could be stated that d109 has all five IE genes deleted and although it enters the nucleus, there is no viral gene expression. Deletion of ICP4 restricts viral gene expression to IE genes. Line 195, hp66 harbors a mutation in the viral DNA polymerase gene.

Response: All references to mutant HSV-1 viruses in the text and figures have been changed to the format of Δ protein or WT, to aid readers. See above our response to reviewer 2 comment #3.

2. In a number of instances, especially in the Discussion, the authors pose a hypothesis to explain their findings but there is no direct evidence presented here and thus they state that "further work needs to be done". A discussion should include speculation and point to future studies but in the end there is a lot of speculation and it is difficult to come up with firm conclusions.

Response: We have revised the discussion section to address the reviewer's comment.

3. There is a confusing section in the Discussion that appears to mix up mRNA and protein. Starting on line 395, the authors describe that productive viral infection takes a toll on host cell homeostasis ... including the unfolded protein response UPR. Line 398 it is stated that "HSV-1 transcripts are composed from a high density of intrinsically disordered domains and due to rapid rate of synthesis are prone to misfolding and aggregation traits that robustly trigger a UPR. Proteins not mRNA transcripts can have intrinsically disordered domains and can under some conditions lead to misfolding and aggregation and this can lead to the Unfolded Protein Response.

Response: We have revised the discussion section to address the reviewer's comment.

4. Lines 418 ad 419, there was no evidence presented in this study for a putative Pol III transcript derived from the HSV-1.

Response: We have rephrased this sentence to the following: "Herein we provide evidence suggesting a previously unrecognized role for Pol III in HSV-1 gene expression."

5. I would suggest the discussion be tightened as it somewhat rambles among the different findings that were observed and the possible hypotheses to explain the findings, to discounting several of these or stating that more work is needed.

Response: We have revised the discussion section to address the reviewer's comment.

Reviewer #1 (Remarks to the Author):

The authors addresses most of my comments satisfactorily, however there are a few minor points that require clarification/correction:

- line 182f: "Based on these results it is unlikely that tRNA upregulation is due to Pol II-transcriptional run off or -transcriptional interference" -> this is based on a location analysis of up- and down-regulated RNAs. However, when evaluating how frequent tRNAs are downstream of genes, they only consider very immediately downstream tRNAs, i.e. within 300bp (unless this is supposed to be 300kb, in which case, please adjust the legend). Everything else is distal intergenic. The authors should include a larger downstream region to exclude that upregulation of tRNAs is in any way associated with Pol II transcriptional read-through observed in HSV-1 infection.

- line 206f: "We employed ERCC spike-in controls to normalize expression relative to rRNA—which remain steady during HSV-1 productive infection" -> ERCC spike-ins were applied to Ribo-depleted total RNA-seq, so this is a bit confusing. I presume they mean to say that since they added spike-ins to total RNA, which consists almost completely of rRNA, before depletion, any change in total RNA resulting from alterations in non-rRNA levels is probably negligible and thus this normalization is appropriate. This sentence should be adjusted to make this clear.

- line 258ff: While ERCC spike-in normalized total RNA is appropriate for absolute quantification of transcripts, polyA-RNA-seq is not as this does not include rRNAs but total polyA-RNA levels likely change due to vhs-mediated decay, reduction in host transcription and viral transcription. Thus, the assumption between RPKM normalization, i.e. the overall RNA levels do not change, is likely invalid here. This may explain the discrepancy between polyA-RNA-seq and total RNA-seq for POLR2A at 12 h p.i., with the first showing no change compared to uninfected and the second a strong reduction. This issue should be discussed in the manuscript.

- line 681ff: while the authors now aim to explain their normalization of the ChIP-seq data, this is still cryptic to me. What is Million/Billion IP reads? How did they determine the HSV-1 genome copy number? What is the distinction between input and IP here? Is input the non-IP background? If yes, why is the norm factor the same for all IPs despite IP total reads included in the calculation? From the data and explanation they provide, I cannot reproduce their calculation. This needs to be revised.

- line 992: Fig. 2 legend: "Differential expression analysis (Benjamini-Hochberg)" -> this should be "Differential expression analysis (edgeR, p-values adjusted for multiple testing by Benjamini-Hochberg)" or something similar. Also applies to legend for Supp. Fig. 2.

- line 1038 ff. Legend to Fig. 5: "Intersection analysis for ..." This needs more explanation. After some thinking, I think I now understood it but for most readers this will be rather cryptic, in particular since the overlap to the factor itself is always also shown. It should also be clarified that in this case, the overlap is always the number of binding sites and the Jaccard statistics is 1, i.e. the maximum possible value. This also applies to similar figures in Fig. 6 and Suppl. Fig 14, but in this case a reference to legend of Fig. 5 is sufficient.

- Fig. 5D and Supp. Fig. 12 and 13: the position of the POLR3A peak should be marked for clarification, e.g. by a rectangle to make it easier for the reader to identify where it should be in cases where it is absent

- line 325: delete the comma after promoters

Reviewer #2 (Remarks to the Author):

My concerns have been fully addressed.

Reviewer #3 (Remarks to the Author):

The authors have addressed all of my concerns and present a clearer description of what was done, what the major findings are and what the the major advances are that were found in these studies. I have no further concerns or comments.

REVIEWER COMMENTS

Reviewer #1 (Remarks to the Author):

The authors addresses most of my comments satisfactorily, however there are a few minor points that require clarification/correction:

1) - line 182f: "Based on these results it is unlikely that tRNA upregulation is due to Pol II-transcriptional run off or -transcriptional interference" -> this is based on a location analysis of up- and down-regulated RNAs. However, when evaluating how frequent tRNAs are downstream of genes, they only consider very immediately downstream tRNAs, i.e. within 300bp (unless this is supposed to be 300kb, in which case, please adjust the legend). Everything else is distal intergenic. The authors should include a larger downstream region to exclude that upregulation of tRNAs is in any way associated with Pol II transcriptional read-through observed in HSV-1 infection.

Response: The labeling is correct. It is 300bp. The Dolken group showed that ICP27 was required for the transcriptional readthrough that the reviewer refers to. Since we found that activation of tRNA transcription does not require ICP27, we didn't believe further analysis was warranted.

2) - line 206f: "We employed ERCC spike-in controls to normalize expression relative to rRNA—which remain steady during HSV-1 productive infection" -> ERCC spike-ins were applied to Ribo-depleted total RNA-seq, so this is a bit confusing. I presume they mean to say that since they added spike-ins to total RNA, which consists almost completely of rRNA, before depletion, any change in total RNA resulting from alterations in non-rRNA levels is probably negligible and thus this normalization is appropriate. This sentence should be adjusted to make this clear.

Response: We have updated line 206 to the following: "We added ERCC spike-in controls to total RNA prior to ribosomal RNA depletion allowing us to normalize expression relative to rRNA which remain steady during HSV-1 productive infection. This method ensures a quantitative comparison of samples with varying levels of host shut off."

3) - line 258ff: While ERCC spike-in normalized total RNA is appropriate for absolute quantification of transcripts, polyA-RNA-seq is not as this does not include rRNAs but and total polyA-RNA levels likely change due to vhs-mediated decay, reduction in host transcription and viral transcription. Thus, the assumption between RPKM normalization, i.e. the overall RNA levels do not change, is likely invalid here. This may explain the discrepancy between polyA-RNA-seq and total RNA-seq for POLR2A at 12 h p.i., with the first showing no change compared to uninfected and the second a strong reduction. This issue should be discussed in the manuscript.

Response: We have adjusted the text to note that RiboMinus RNA-Seq is the more quantitative comparison. While we pointed out that POLR2A levels increased according to PolyA-Seq, both Total RNA-Seq and protein levels (Sup Fig. 10) show a decrease which we also note in the text.

4) - line 681ff: while the authors now aim to explain their normalization of the ChIP-seq data, this is still cryptic to me. What is Million/Billion IP reads? How did they determine the HSV-1 genome copy number? What is the distinction between input and IP here? Is input the non-IP background? If yes, why is the norm factor the same for all IPs despite IP total reads included in the calculation? From the data and explanation they provide, I cannot reproduce their calculation. This needs to be revised.

Response: We have added the following text at lines 597-598, 672-674, 692-693 to clarify what input/IP samples means:

"Input samples were taken after chromatin was sheared by sonication and prior to antibody-based immunoprecipitation."

"For every infection performed we sequenced a paired input sample to determine the relative contributions of host and HSV-1 genome content in all conditions."

Where “Input” is the paired chromatin sample before immunoprecipitation and “IP” is the immunoprecipitated (i.e. POLR3A, POLR2A, etc.) chromatin sample.”

The norm factor is not the same for all IP’s, the representative data shown is all for the different input samples sequenced. We realized that the top row was labeled as “IP” which was likely confusing, considering all samples shown are actually Input. We have updated the table information to clarify this.

We have also added in text to aid in calculating viral genome copy number at lines 681-687.

5) - line 992: Fig. 2 legend: "Differential expression analysis (Benjamini-Hochberg)" -> this should be "Differential expression analysis (edgeR, p-values adjusted for multiple testing by Benjamini-Hochberg)" or something similar. Also applies to legend for Supp. Fig. 2.

Response: We have updated the text of the legend for Fig. 2 and Sup Fig. 2 as suggested.

6) - line 1038 ff. Legend to Fig. 5: "Intersection analysis for ..." This needs more explanation. After some thinking, I think I now understood it but for most readers this will be rather cryptic, in particular since the overlap to the factor itself is always also shown. It should also be clarified that in this case, the overlap is always the number of binding sites and the Jaccard statistics is 1, i.e. the maximum possible value. This also applies to similar figures in Fig. 6 and Suppl. Fig 14, but in this case a reference to legend of Fig. 5 is sufficient.

Response: We have added the following text to the legends of Fig. 5, 6 and Sup Fig. 14 to clarify the statistical comparison:

“Intersection size is the number of instances where peaks overlap by at least 1 nucleotide. Jaccard statistic is the degree of overlap between all peak regions, with the maximum value being 1.”

7) - Fig. 5D and Supp. Fig. 12 and 13: the position of the POLR3A peak should be marked for clarification, e.g. by a rectangle to make it easier for the reader to identify where it should be in cases where it is absent

Response: We understand it may be difficult to see the exact location where POLR2A and POLR3A overlap or bind uniquely—which is why we statistically analyzed the intersection of POLR3A and POLR2A in Fig. 5A-B, and also show overlaid and separated traces for the two factors. Due to the density of factor binding and because we have already provided the information in Fig. 5A-B, we do not think the addition of boxes in figure 5D will be helpful.

8) - line 325: delete the comma after promoters

Response: This comma was deleted.

Reviewer #2 (Remarks to the Author):

My concerns have been fully addressed.

Reviewer #3 (Remarks to the Author):

The authors have addressed all of my concerns and present a clearer description of what was done, what the major findings are and what the major advances are that were found in these studies. I have no further concerns or comments.

Peer review comments, final comments

Reviewer #1 (Remarks to the Author):

My concerns have been addressed.